# The MOSAiC ice floe: sediment-laden survivor from the Siberian shelf

Thomas Krumpen[1], Florent Birrien[1], Frank Kauker[1], Thomas Rackow[1], Luisa von Albedyll[1], Michael Angelopoulos[1], H. Jakob Belter[1], Vladimir Bessonov[2], Ellen Damm[1], Klaus Dethloff[1], Jari Haapala[3], Christian Haas[1], Carolynn Harris[4], Stefan Hendricks[1], Jens Hoelemann[1], Mario Hoppmann[1], Lars Kaleschke[1], Michael Karcher[1], Nikolai Kolabutin[2], Ruibo Lei[5], Josefine Lenz[1,6], Anne Morgenstern[1], Marcel Nicolaus[1], Uwe Nixdorf[1], Tomash Petrovsky[2], Benjamin Rabe[1], Lasse Rabenstein[7], Markus Rex[1], Robert Ricker[1], Jan Rohde[1], Egor Shimanchuk[2], Suman Singha[8], Vasily Smolyanitsky[2], Vladimir Sokolov[2], Tim Stanton[9], Anna Timofeeva[2], Michel Tsamados[10], Daniel Watkins[11]

[1]Alfred Wegener Institute, Helmholtz Centre for Polar and Marine Research, Am Handelshafen 12, 27570 Bremerhaven, Germany
[2]Arctic and Antarctic Research Institute, Ulitsa Beringa, 38, Saint Petersburg, 199397, Russia
[3]Finnish Meteorological Institute, Marine Research, P.O.Box 503, FI-00101 Helsinki, Finland
[4]Dartmouth College, Department of Earth Science, 6105 Fairchild Hall, Hanover, NH, 03755, US
[5]Polar Research Institute of China, MNR Key Laboratory for Polar Science, 451 Jinqiao Road, Pudong, Shanghai 200136, China
[6]Association of Polar Early Career Scientists, Alfred Wegener Institute for Polar and Marine Research, Telegrafenberg A45, 14473 Potsdam, Germany
[7]Drift & Noise Polar Services, Stavendamm 17, 28195 Bremen, Germany
[8]German Aerospace Center, Remote Sensing Technology Institute, SAR Signal Processing, Am Fallturm 9, 28359 Bremen, Germany
[9]Naval Postgraduate School, Oceanography Department, 833 Dyer Road, Building 232, Monterey, CA 93943, US
[10]Centre for Polar Observation and Modelling, University College London, Dept. of Earth Science, 5 Gower Place, London WC1E 6BS, UK
[11]College of Earth, Ocean, and Atmospheric Science, Oregon State University, Corvallis, OR, US

*Correspondence to*: Thomas Krumpen (tkrumpen@awi.de)

**Abstract.** In September 2019, the research icebreaker *Polarstern* started the largest multidisciplinary Arctic expedition to date, the MOSAiC (Multidisciplinary drifting Observatory for the Study of Arctic Climate) drift experiment. Being moored to an ice floe for a whole year, thus including the winter season, the declared goal of the expedition is to better understand and quantify relevant processes within the atmosphere-ice-ocean system that impact the sea ice mass and energy budget, ultimately leading to much improved climate models. Satellite observations, atmospheric reanalysis data, and readings from a nearby meteorological station indicate that the interplay of high ice export in late winter and exceptionally high air temperatures resulted in the longest ice-free summer period since reliable instrumental records began. We show, using a Lagrangian tracking tool and a thermodynamic sea ice model, that the MOSAiC floe carrying the Central Observatory (CO) formed in a polynya event north of the New Siberian Islands at the beginning of December 2018. The results further indicate that sea ice in the vicinity of the CO (< 40 km distance) was younger and 36 % thinner than the surrounding ice with potential consequences for ice dynamics and momentum and heat transfer between ocean and atmosphere. Sea ice surveys carried out on various reference floes in autumn 2019 verify this gradient in ice thickness, and sediments discovered in ice cores (so called dirty sea ice) around the CO confirm contact with shallow waters in an early phase of growth, consistent with the tracking analysis. Since less and less ice from the Siberian shelves survives its first summer (Krumpen et al., 2019), the MOSAiC experiment provides the unique opportunity to study the role of sea ice as a transport medium for gases, macro-nutrients, iron, organic matter, sediments, and pollutants from shelf areas to the central Arctic Ocean and beyond. Compared to data for the past 26 years, the

sea ice encountered at the end of September 2019 can be already classified as exceptionally thin, and further predicted changes towards a seasonally ice-free ocean will likely cut off the long-range transport of ice-rafted materials by the Transpolar Drift in the future. A reduced long-range transport of sea ice would have strong implications for the redistribution of biogeochemical

matter in the central Arctic Ocean, with consequences for the balance of climate relevant trace gases, primary production and biodiversity in the Arctic Ocean.

**1 Introduction**

In early autumn 2019 the German research icebreaker *Polarstern*, operated by the Alfred Wegener Institute (AWI), Helmholtz Centre for Polar and Marine Research, was moored to an ice floe north of the Laptev Sea in order to travel with the Transpolar Drift on a one-year long journey toward Fram Strait. The goal of the international Multidisciplinary drifting Observatory for the Study of Arctic Climate (MOSAiC) project is to better quantify relevant processes within the atmosphere-ice-ocean system that impact the sea ice mass and energy budget. Other main goals are a better understanding of available satellite data via

ground-truthing and improved process understanding that can be implemented into climate models. MOSAiC continues a long tradition of Russian North Pole (NP) drifting ice stations. In the past, these stations predominantly used older multi-year ice floes as their base of operations, with small settlements set up on the surface. Using this approach, the Arctic and Antarctic Research Institute (AARI, Russia) undertook 40 NP drift stations in the Central Arctic between 1937 and 2013. However, as the summer melt period lasted longer every year, thick multi-year floes suitable for ice camps became more seldom, and Russia

was ultimately forced to temporarily discontinue these drifting stations.

The MOSAiC project represents an attempt to adapt to the "new normal" in the Arctic (warmer and thinner Arctic sea ice) and to use the ship itself as an observational platform. Around the ship, an ice camp (Central Observatory, CO) with comprehensive instrumentation was set up to intensively observe processes within the atmosphere, ice, and ocean. For this purpose, on October

4, 2019, the ship was moored to a promising ice floe measuring roughly 2.8 x 3.8 km (see Fig. 1 at coordinates 136°E, 85°N). The floe was part of a loose assembly of pack ice, not yet a year old, which had survived the summer melt (hereafter called residual ice (WMO 2017), shorthand for residual first year ice, which does not graduate to become second year ice until January 1$^{st}$). With the support of the Russian research vessel *Akademik Fedorov*, a Distributed Network (DN) of autonomous buoys was installed in a 40-km radius around the CO on 55 additional residual ice floes of similar age. For more information

about the MOSAiC expedition the reader is referred to www.mosaic-expedition.org.

The purpose of this paper is to investigate the environmental conditions that shaped the ice in the chosen research region prior to and at the start of the MOSAiC drift. The analyses presented here are of high importance for future work as they will provide the initial state for model-based studies and satellite-based validation planned to take place during MOSAiC. In addition, it

provides the foundation for the analysis and interpretation of upcoming biogeochemical and ecological studies. This study exclusively employs previously described methods (Damm et al., 2018, Peeken et al., 2018, Krumpen et al., 2016, Krumpen et al. 2019) for tracking sea ice back in time, and for modeling thermodynamic sea ice evolution (see Methods). These tools are used in combination with the first field observations made on board the accompanying research vessel *Akademik Fedorov*. A more detailed description of the CO's physical characteristics will be the focus of future studies.


We first provide an overview of the ice conditions in the extended surroundings of the experiment, and of the atmospheric and oceanographic processes that preconditioned the ice in the preceding winter and summer. To do so, we utilise satellite observations, NCEP atmospheric reanalysis data, and readings from a nearby meteorological station.

Secondly, we evaluate the representativeness of the ice conditions in *Polarstern*'s immediate vicinity compared to the extended surroundings. These analyses chiefly employ a Lagrangian backward tracking tool (see Methods) that allows us to determine where the encountered ice was initially formed, and to identify the dominant processes that have influenced the ice along its trajectory. For this work, a thermodynamic one-column model was coupled to the backtracking tool to simulate ice growth and melting processes along these trajectories (Methods). The coupled results are then compared with observational data

gathered by satellites and in-situ measurements made during the search for the main floe and setup of the DN.

Thirdly, we discuss whether the ice encountered in autumn 2019 on-site was unusually thin compared to previous years. For this we run the coupled thermodynamics-tracking model for the MOSAiC start region with NCEP forcing data of the past 26 years to examine interannual variability of residual ice thickness in the study region.

In closing, implications for upcoming future physical, biogeochemical, and ecological MOSAiC studies due to the conditions encountered on site are discussed.

## 2 Material and Methods

### 2.1 Lagrangian sea ice trajectories

To determine the origin, pathways and thickness changes of sea ice, as well as the atmospheric forcing acting on the ice cover, we use our Lagrangian drift analysis system called IceTrack that traces sea ice backward in time using a combination of satellite-derived, low-resolution drift products (Krumpen et al., 2019). The approach has also been applied in a number of previous studies for the same purpose (Ricker et al., 2018, Damm et al., 2018, Peeken et al., 2018, Krumpen et al., 2016 and others). In summary, IceTrack uses a combination of three different, publicly available ice drift products for the tracking: i)

motion estimates based on a combination of scatterometer and radiometer data provided by the Center for Satellite Exploitation and Research (CERSAT, Girard-Ardhuin et al., 2012), ii) the OSI-405-c motion product from the Ocean and Sea Ice Satellite Application Facility (OSI SAF, Lavergne et al., 2016), and iii) Polar Pathfinder Daily Motion Vectors (v.4) from the National Snow and Ice Data Center (NSIDC, Tschudi et al., 2016). The contributions of individual products to the used motion field are weighted based on their accuracies and availability which vary with seasons, years, and study region. The IceTrack

algorithm first checks for the availability of CERSAT motion data within a predefined search range. CERSAT provides the most consistent time series of motion vectors starting from 1991 to present and has shown good performance on the Siberian shelves (Rozman et al. 2011). During summer months (June–August) when drift estimates from CERSAT are missing, motion information is bridged with OSISAF (2012 to present). Prior 2012, or if no valid OSISAF motion vector is available within the search range, NSIDC data is applied. The tracking approach works as follows: Ice in user-defined individual starting

locations or positions on a 25 km EASE2 grid is traced backward in time on a daily basis. Tracking is discontinued if a) the tracked ice reaches the coastline or fast ice edge, or b) the ice concentration at a specific location along the backward trajectory drops below 40 % and we assume the ice to be formed.

### 2.2 Auxiliary data extracted along the track:

#### 2.2.1 Ice concentration and water depth

Ice concentration along the trajectories is provided by CERSAT and based on 85 GHz SSM/I brightness temperatures. The CERSAT product makes use of the ARTIST Sea Ice (ASI) algorithm and is available on a 12.5 km × 12.5 km grid (Ezraty et al., 2007). Information on water depth was obtained from the International Bathymetry Chart of the Arctic Ocean (IBCAO, Jakobsson et al., 2012).


#### 2.2.2 Satellite-based and model-based sea ice thickness estimates

The satellite-based sea ice thickness observations used in this study are based on the weekly merged CryoSa-2/SMOS sea ice thickness product provided on a 25 km EASE2 grid by the AWI (Ricker et al., 2017). Weekly estimates from April were then averaged in order to obtain monthly sea ice thickness estimates for April 2019 (compare Fig. 2a).


In addition to satellite-based mean thickness estimates, the level ice thickness was computed along the Lagrangian drift trajectories by means of the one-dimensional thermodynamic model Icepack (cf. CICE Consortium) that drifted with the ice. The single-column model describes the seasonal evolution of thickness distribution for a single floe from an initial ice thickness. It uses an approach combining seven ice categories and seven layers (only 1 layer of snow), and accounts for

thermodynamic growth and melting as well as mechanical redistributions due to ridging (e.g. Thorndike et al., 1975, Lipscomb et al., 2001). For the purpose of this study, the mechanical aspect was disregarded in order to focus on thermodynamically grown level ice. At each time step, the growth and melt rates are derived from heat fluxes based on atmospheric and oceanic forcing by solving conservation laws of snow and ice enthalpy (e.g. Bitz et al., 1999). Every simulation began with open ocean

conditions. The atmospheric forcing was provided by NCEP reanalysis data (Kanamitsu et al., 2002) and consisted of
downward short- and longwave radiation fluxes, the surface air temperature and specific humidity, wind field and precipitation.
The oceanic forcing, including sea surface temperature and salinity, was derived from a climatology based on hydrographic
surveys carried out in the Laptev Sea (Janout et al., 2016), where most of the ice originated.

## 2.3 Area flux estimates

To investigate the impact of winter sea ice dynamics on the summer ice cover, we calculate monthly sea ice area fluxes through
the northern boundary of the Laptev Sea for the winter season from March to April (1992 – 2019). The gate is located between
110°E and 160°E at 77.5°N (black line with arrows in Fig. 2a). The flux calculations follow the approach of Ricker et al.
(2018) who estimated volume fluxes through Fram Strait. For ice concentration, we use the CERSAT product. For ice motion,
we use merged products from CERSAT that are based on radiometer and scatterometer data. Figure 2c shows the total ice area
export from March – April of each winter, including a trend line plotted on top.

## 2.4 Sea ice break-up and freeze-up

The timing of sea ice break-up and freeze-up (Fig. 2b) was estimated for each year based on CERSAT sea ice concentration
data for the region between 86°N, 100°E and 71°N, 160°E. An ice free grid point is defined as the first day in a series of at
least 10 days when ice concentration exceeds and reaches zero, respectively (Janout et al., 2016).

## 2.5 Field observations:

### 2.5.1 Snow and ice thickness measurements

Ground-based electromagnetic (GEM) induction measurements of ice thickness were obtained on five different residual ice
floes between October 1 and October 7: Four floes were located in the vicinity of the CO (~ 15 km) and part of the DN (see
Fig. 3a, L1-L3, M8). The fifth floe was positioned outside the DN and will hereafter be called Reference Site R1.

The GEM was mounted on a plastic sledge and pulled across the snow surface. The most frequently occurring ice thickness,
the mode of the distribution (compare Fig. 6), represents level ice thickness and is the result of winter accretion and summer
ablation. According to Haas and Eicken (2001), a comparison of GEM measurements performed in the Central Arctic during
summer month with drill hole data indicate that the accuracy of the induction measurements is better than 0.05-0.10 m and
that the method is well suited for high-resolution thickness profiling. For further details on the data processing and handling
we refer Hunkeler et al. (2016).

It is important to note here that electromagnetic sounding only yields the total ice thickness (snow thickness plus sea ice
thickness). Therefore the snow surface layer thickness has to be to be measured independently to yield ice thickness. Snow
thickness measurements on L1-L3 and M8 were obtained every 2 – 5 m along the GEM tracks with a Magna Probe (Snow
Hydro, Fairbanks, AK, USA). At R1, manual snow thickness measurements were taken at randomly selected locations. After
GEM and Magna Probe measurements were converted to a drift- and rotation-corrected coordinate system using a GPS reference
station, sea ice thickness was calculated by subtracting total ice thickness from snow thickness.

While searching for a suitable floe for the CO, two additional regions were visited (see Fig. 3a, R2 and R3), each consisting
of a collection of smaller floes. Here, manual ice and snow thickness measurements were taken on the level ice with a drill,
measuring stick, and thickness gauge.

Table 1 summarises the mean and modal thickness of sea ice and snow for all individual sampling sites.

### 2.5.2 Ice coring

Ice cores were taken at all the L sites (Fig. 3a) with a standard 9 cm Kovacs ice corer. At L1, four cores were collected. At L2,
three cores were taken from level ice and three cores from a ridge at different surface elevations. At L3, three cores were
extracted from level ice and three cores at the lower relief area of a ridge. Within the MOSAiC central floe, ice coring took
place at several sites on a weekly basis, but only the sediment-laden sea ice observed at one of the residual ice stations is
discussed in this paper. The ice cores were sectioned into 10 cm samples, melted, and then filtered for sediments using 0.45
μm filters. At all sampling sites, parallel cores were taken and stored at -20 °C for future methane concentration and isotope
analysis. Since the MOSAiC floes may originate from methane super-saturated seawater near the Siberian coast, some of the
residual ice may contain relict biogeochemical conditions from the initial ice formation. This further demonstrates the
importance of understanding the history of the MOSAiC floe for future studies.

### 2.5.3 Ice observations from the bridge

On board of *Akademik Fedorov*, visual ice observations were carried out from the bridge by a group of three specially trained
ice observers.  Detailed descriptions of the methodology and protocols applied are provided in Alekseeva et al. (2019) and
AARI (2011), all congruent to the WMO Sea Ice Nomenclature (2017). Continuous 24-hour ice observation were available
from September 28 (approaching R1) to October 3 (approaching the DN). The observations included visual descriptions of the
ice cover's main characteristics, i.e. total concentration and partial concentrations and forms of the encountered stages of ice
development, hummock and ridges concentration, melting stage, and the sizes and orientations of fractures and leads. In this
paper, we will use the observed (within the limits of horizontal visibility) residual ice fraction along the ship's track (see Fig.
5). Data was resampled to an hourly interval.

## 3 Results and Discussion

**3.1 Sea ice retreat in summer 2019: Preconditioning processes**

Sea ice retreat during the melting period in the Laptev Sea and East Siberian Sea is the result of atmospheric and oceanic
processes and regional feedback mechanisms acting on the ice cover, both in winter and summer. In the following, we will
briefly review the sea ice conditions on the Siberian Shelf seas prior to the start of the expedition and the main preconditioning
mechanisms that contributed to the northward retreat of the ice edge in 2019.  In this regard, our focus is on the atmospherically
driven processes, since results from oceanographic surveys are not yet available.

Ice dynamics and ice export in winter are important preconditioning mechanisms for the ice retreat in summer. Itkin and
Krumpen (2017) observed that enhanced offshore-directed transport of sea ice in late winter has a thinning effect on the ice
cover. During late winter months dominated by an offshore-directed drift component, newly formed ice areas are larger and
remain comparatively thin and therefore melt more rapidly once temperatures rise above freezing. This feedback mechanism
is even more pronounced when temperatures at the end of winter are unusually high. Figure 2 summarises the conditions and
processes that shaped ice formation in the Laptev Sea and East Siberian Sea in winter 2018/2019. Satellite-based estimates of
offshore-directed sea ice area transport between March and April are shown in Fig. 2c (1992 – 2019, from 110°E to 160°E at
77.5°N). Late winter flux estimates indicate that the sea ice advection away from the Siberian Shelves towards the Central
Arctic was approximately 70 % higher ($2.32 \times 10^5$ km²) in 2019 than the long-term mean annual rate (~$1.36 \times 10^5$ km²).

Following Krumpen et al. (2013), the strong positive trend (+0.53 x $10^5$ km² / decade) in late winter ice area export is associated with an increasing drift speed as a result of thinning ice cover and a rapid loss of thick multi-year ice. As a consequence of the intensified ice advection shortly before spring break, satellite-based sea ice thickness observations (Fig. 2a) show negative thickness anomalies throughout the entire coastal zones of the East Siberian Sea and the Laptev Sea in April 2019, except for southern half of the area around the New Siberian Islands.

Ocean-driven preconditioning mechanisms are less well understood. However, there is indication that enhanced winter ventilation of the ocean can reduce sea ice formation in this area at a rate now comparable to losses from atmospheric thermodynamic forcing (Polyakov et al., 2017). Observations carried out in the eastern Eurasian Basin have shown that weakening of the halocline and shoaling of intermediate-depth Atlantic water layer results in heat flux equivalent to 40–54 cm reductions in ice growth in 2013/2014 and 2014/2015.

In addition, anomalously high temperatures during the winter months can further reduce the growth of first-year ice (FYI), resulting in thinner ice cover at the end of the winter (Ricker et al., 2017). According to NCEP reanalysis data (Fig. A, Supplement) and observations from the Kotelny meteorological station (Fig. 2a, yellow circle), the temperatures during the ice growth phase (October 2018 – May 2019) were elevated: reanalysis data show positive temperature anomalies of 3 °C in comparison to the 1981 - 2010 climatology, and records at Kotelny show significantly higher temperatures than those at the beginning of the instrumental record (Fig. 2e). In particular, temperatures at the end of the winter are unusually high. If this coincides, as described above, with periods of strong offshore-directed winds, the formation of new ice in coastal areas is reduced, which favours early melting of the ice cover in spring (Fig. B, Supplement).

The subsequent temperature anomalies in spring and summer 2019 were even more pronounced. During the summer months, Kotelny meteorological monitoring station recorded the highest mean temperatures since the beginning of record-keeping (Fig. 2d), and the reanalysis data indicates a positive anomaly of 2.5 degrees on the Siberian Shelves and in adjacent northern regions (Fig. A, Supplement). The rapidly rising temperatures in spring accelerated the melting of the ice cover, which was extremely thin to begin with (Fig. 2a). This resulted in the earliest ice break-up ever observed (compare Fig. 2b, red line) and rapid northward retreat of the ice edge, which exposed surface waters to direct solar heating. Consequently, summer (August 2019) sea surface temperatures south of the MOSAiC starting area were approximately 2-4°C higher than the 1982 - 2010 mean (Timmermans and Ladd, 2019), such that wind events that force ice floes back into warm waters could have caused additional ice melt (Steele and Ermold, 2015). Moreover, the intensive warming of the upper ocean (Janout et al., 2016) caused a delay in the autumnal freeze-up of sea ice (Fig. 2b, blue line) and resulted in large parts of the marginal seas remaining ice-free for up to 93 days. This means that the MOSAiC expedition started immediately after the longest recorded ice-free period in the region.

## 3.2 Sea ice origin and initial conditions in September 2019

In this section we describe the predominant ice conditions at the beginning of MOSAiC, both in the ship's immediate vicinity and its extended surroundings. The latter encompass the area within a 220-km radius of *Polarstern* and will hereafter be referred to as the Extended MOSAiC Region (EMR, see Fig. 3a). A radius was selected to include various ice types, which differ in terms of their provenance (i.e. origin) and/or age. The EMR includes both the ice edge to the south, and thicker and more stable pack ice to the north. The ship's immediate vicinity (Distributed Network Region, DNR) includes the DN and has a radius of 40 km. We will first describe the ice conditions in the EMR, before turning our attention to the DNR.

Once the MOSAiC floe had been chosen, we applied a tracking tool (see Methods) to the residual ice that was in the EMR shortly before MOSAiC's starting date. Figure 3b shows the age of the sea ice within the EMR on September 25. Based on the backtracking analysis, the EMR's residual ice had an average age of 318 days, and was formed on November 11, 2018 (+/- 15 days). Second-year (SYI) or multi-year ice (MYI) was not found, neither from tracking nor from scatterometer data. Most of the residual ice was originally produced during or shortly after the freeze-up in polynyas (or elsewhere on the shallow Siberian Shelves) (Fig. 3c), featuring water depths of less than 30 m. Only the ice at the far eastern and northern edges of the EMR originated from regions with a water depth exceeding 50 m. From the time of its formation to September 25, the EMR ice had travelled an average distance of approximately 2440 km (+/- 205 km, Fig. 3d), and experienced low ice concentrations between June and September 2019 (Fig. 3e). Hence, the residual ice encountered after our arrival on site was severely weathered, and bridge observations indicated that a large fraction was melted completely during summer months. Residual ice that survived was characterised by frozen-over melt ponds with a <10 cm deep layer of fresh snow. Based on visual observation, melt pond fraction was 70 - 80 % in the undeformed ice areas and the bottom layer experienced internal melting. According to ice coring, only the top 30 centimetres of ice was solid. Because both ships only reached the target region after the freeze-up had begun, large expanses of previously open water were now covered with new ice.

Based on the backtracking analysis, the floes selected for the Central Observatory and the DN were located in a zone of comparatively young ice that formed roughly three weeks later than the ice within the EMR (Fig. 3b, early December 2018) and originated from a shallow (Fig. 3c) region closer to its location on September 25th (Fig. 3d, 2240 km). Figure 4a shows the trajectories obtained for the centre of the DNR (the position of the CO, red line) and four adjacent positions at a distance of 25 km (grey lines). Information on water depths and ice concentration along the central trajectory is provided in Fig. 4b/c. The trajectories indicate that the ice inside the DNR was formed in a polynya event on December 5, 2018, north of the New Siberian Islands in water that was less than 10 m deep. An eastward ice drift then transported the newly formed ice along the shallow shelf, until it reached deeper water in early February 2019. Ice cores collected at various points in the DN and on the CO confirm that the DNR ice originated in the shallow Siberian Shelves, since some of the cores contained sediment inclusions of sandy silt in the uppermost 50 cm (Fig. 4c/d). Though the quantities were small in most cases, these inclusions can only be found on the shallow Siberian Arctic Shelves with average water depths of less than 30 m (Sherwood et al. 2000, Wegener et al. 2017). There, particulate matter and organisms are incorporated into the newly formed ice by suspension freezing (Eicken et al. 2000) or, to a smaller degree, by grounded sea ice pressure ridges ploughing through the seafloor (Darby et al. 2011). A detailed chemical analysis of these trapped sediments will be conducted at a later point in time.

The validity and reliability of Lagrangian drift studies depend on the accuracy of the applied sea ice motion product. In this study, we primarily use the CERSAT drift dataset because it provides the most consistent time series of motion vectors starting from 1991 to present (see Methods). Comparisons with buoys and high resolution SAR images indicate that in particular during winter months, when the atmospheric moisture content is low and surface melt processes are absent, the quality of motion products from low resolution satellites is high (Sumata et al. 2014, Krumpen et al. 2019). Restrictions may arise from the coarse resolution of the sensors in near-shore regions characterized by a complex coastline, extensive fast ice areas, and polynyas (Rozman et al. 2011). During summer months (June – August), when strong surface melt processes and high moisture content in the atmosphere further reduces accuracy of low resolution motion products (Sumata et al. 2014), IceTrack uses the OSISAF motion product to bridge the lack of CERSAT data. To quantify uncertainties of sea ice trajectories on a larger temporal and spatial scale, we reconstructed the pathways of drifting buoys using IceTrack. For this purpose, we selected 10 buoys that had survived a full summer and winter in the Arctic. Their drift was then reproduced from October onwards in a backward direction over 12 months. Fig. C (Appendix) shows the deviation between actual and virtual tracks, which is rather small (60 +/-24 km after 320 days) and in an acceptable range. The maximum deviation between real and virtual buoys gives

a measure of the largest possible error that can occur when determining the ice origin. After 320 days it is around 105 km. The confidence bound is shown in Fig. 4 as an ellipsoid (dashed line). No significant differences in sea ice pathways and source areas were observed when repeating the tracking experiment using different combinations of motion products, or higher/lower ice concentration thresholds.


Note that we originally planned to trace the provenance of the MOSAiC floe using high-resolution satellite data (Sentinel-1, TerraSAR-X, and MODIS). However, only sporadic high-resolution images of the region were available, and the combination of low summertime sea ice concentration and high degree of cloud cover made it extremely difficult to manually track the exact position of individual floes over an extended period of time. Nevertheless, the high resolution satellite data enabled us

to track nearby large scale patterns such as shear zones or very prominent floes. Hence, we could at least determine the approximate location of the MOSAiC floe on individual images. The resulting estimates for the different positions of the CO (brown-yellow colored circles in Fig. 4a) correspond well to the computed trajectories (red line in Fig. 4a), which lends increased confidence in our results.

To calculate the ice thickness variability in the EMR and DNR at the start of MOSAiC (Fig. 5 a), and the ice thickness evolution along the drift trajectories encountered by the ice in those regions (Fig. 5b), we used the results of a thermodynamic model (see Methods). Results show that the residual ice in the DNR was not only younger and originated from a different location than the ice in the surrounding EMR, but it was also thinner: On September 25, the averaged ice thickness inside the EMR was 0.58 m (+/- 0.27 m), while the thickness of ice inside the DNR was 0.37 (+/- 0.09 m), i.e. 36 % (0.21 m) less than in the EMR.

To confirm model results, we applied a second, simpler thermodynamic model developed by Thorndike et al. (1992) and used in Peeken et al. (2018) and Krumpen et al. (2019). The model is chiefly based on air temperatures, assumes a constant ocean heat flux, and employs snow climatology, but indicates the existence of similar thickness gradients between the EMR and DNR (40 % difference, results not shown here). Nevertheless, the decrease in ice thickness toward the DNR is clearly recognisable in both models, and, is in agreement with direct field observations: Figure 6 shows the results of the GEM ice

thickness measurements carried out on four floes in the Distributed Network (L1-L3 and M8), and compares them with measurements taken on R1. The measured difference in modal ice thicknesses (without snow) between R1 and the DNR was 0.3 m (R1: 0.5 m vs. DNR: 0.2 m). Higher ice thicknesses were also measured at R2 and R3 located farther to the north and west, which were reached by helicopter (Table 1, Methods).

Visual observations made from the bridge of the *Akademik Fedorov* as it travelled along the expedition route provided further evidence for the presence of a thickness gradient between the DNR and EMR. The percentage of residual ice steadily dropped from nearly 90 % at R1 to 20 % at the DNR; conversely, the percentage of thin, newly formed ice rose from 10 % to ca. 80 %. This indicates that, given its lower initial thickness at the end of the winter, some of the ice in the DNR could have completely melted in summer. The thickness gradient between the DNR and EMR is confirmed by CryoSat-2/SMOS measurements from

the end of winter 2018/2019. Already in April 2019, a negative thickness anomaly prevails at the later starting position of the drift experiment (Fig. 2a and 3f).

### 3.3 MOSAiC ice conditions compared to previous years

We showed that due to its younger age and different provenance, the DNR ice was thinner than the surrounding ice. But the

thicknesses measurements summarized in Figure 6 and Table 1 are also much smaller than what was observed by Haas and Eicken (2001) in the 1990s by similar GEM and drill-hole measurements. They found late-summer modal FYI thicknesses between 1.25 m (1995), 1.75 m (1993), and 1.85 m (1996) in regions near to or south of the MOSAiC study region, supporting

the notion of exceptionally thin ice in the MOSAiC starting region. In this section, we compare the conditions we encountered at the end of September 2019 with those of previous years by applying the combined tracking-thermodynamics model to the period between 1994 and 2019. Figure 7a shows the history and variation of imaginary MOSAiC floe trajectories for the past 26 years. Tracking was performed backwards in time starting from the DNR region on September 25 of each year. Results indicate that the climatological probability that DNR ice originates from the New Siberian Islands, like in 2019, is about 25% (red shaded area and tracks). From a climatological perspective, it is usually more likely that the ice at the starting position has its origin in the Laptev Sea (55%, light blue shaded area). A smaller part (~20%) typically comes from the East Siberian Sea (grey shaded area). The approximate age of the ice near the starting point is either around one or two years (Fig. 7b), with a tendency towards decreasing ice age. This tendency of decreasing ice age is evident from the frequency of SYI. While SYI occurred in about 64% of all years between 1992 and 2004, it was already much less frequent during the past 15 years (20%, 2005-2019).

Figure 7c displays the time series of September FYI thickness estimates in the DNR for the period between 1994 and 2019. In addition, Fig. 7d provides the annual cycle of DNR ice growth and melt. An overall decrease in residual ice thickness between 1994 and 2019 is visible (trend: -0.22 m/decade), which is subject to a high interannual variability and therefore not statistically significant. The DNR ice encountered in September 2019 can be classified as exceptionally thin over a longer period of time (Fig. 7c). However, for the larger region of the EMR, ice thicknesses in September 2019 agree well with the long-term average (Fig. 7d). Both DNR and EMR ice shows above-average growth rates in winter 2018/2019 as well as above-average thicknesses at the end of April, followed by above average melt. An in-depth analysis of the applied forcing data in the thermodynamic model reveals that the intensified ice production is a consequence of reduced precipitation rates in winter 2018/2019 (Fig. D, Appendix).

Through a comparison with in-situ data, we have shown above that the thermodynamic model is able to simulate regional differences in ice thickness. However, in order to verify that the model is capable to reproduce the interannual variability correctly, model estimates require comparison to historical observational data from the past. Unfortunately, field surveys in this exact location and that time of the year are scarce, but GEM ice thickness measurements in the surroundings of the DNR between 84N and 86.5N, and 100E and 150E (compare Fig 3) were obtained by Haas and Eicken (2001) during the ARK-12 cruise of *Polarstern* in August 1996. The authors obtained around 37 km of thickness profile data at 5 m horizontal spacing. They found average FYI modal thicknesses of ~1.85 m, typical for SYI or even MYI in summer. The 1996 GEM measurements were obtained six weeks earlier in the melt season (August 10 to 22, 1996) inside the EMR area and south of it. In comparison, the exceptionally thick September 1996 ice is reproduced by our thermodynamic model with 1.6 m in the DNR (Fig. 7c). According to Haas and Eicken (2001), the relatively thick ice in 1996 was due to specific atmospheric circulation conditions during summer, characterized by persistent low sea level pressure over the central Arctic. This resulted in very weak surface melt and the absence of melt ponds north of approximately 84°N in 1996. The model results and forcing data for 1996 confirm that strongly reduced net shortwave fluxes led to a significant reduction in ice melting during the summer months. Even in years dominated by strong melting processes, the model seems to realistically reproduce ice thickness: In winter 2013/2014, ice formed comparatively late in the season and melted completely during summer (Fig. 7d). Satellite sea ice concentration data confirm that the DNR region and large parts of the EMR were ice-free already at the beginning of August 2014. If combined with reliable trajectory and realistic forcing data, the good agreement between the thermodynamic model and observations for the years 1996, 2014 and 2019 shows that the model can be used to study interannual variability of FYI thickness changes and the driving mechanisms behind them.

**4 Conclusion and implications for future studies**

In this study, we investigate the initial ice conditions and preconditioning mechanisms at the start of the MOSAiC drift experiment. Moreover, we evaluate how representative the ice within the Distributed Network Region (DNR) is compared to the experiment's extended surroundings (Extended MOSAiC Region, EMR), and question whether the ice encountered was unusually thin compared to past years.

An analysis of satellite-based observations, reanalysis data and readings from the meteorological station Kotelny from 2019 indicates that sea ice retreat in the Siberian Shelf seas was strongly influenced by ice dynamics in late winter and unusually high temperatures in summer. A high offshore directed transport of sea ice shortly before the onset of spring resulted in unusually thin ice cover throughout the entire coastal zones of the marginal seas in April. Rapidly rising temperatures with record temperatures in summer accelerated the melting of the thin ice cover and caused the earliest break-up since 1992.
Intensive warming of the upper ocean further delayed freeze-up and led to the longest ice-free period since the beginning of satellite observations.

Backward trajectories of sea ice present in the large EMR around *Polarstern* during the initial phase of the MOSAiC drift experiment indicate that the majority of residual ice was formed shortly after freeze-up in 2018. In comparison, the ice within
the smaller DNR around *Polarstern* was three weeks younger and formed on the shallow shelves north of the New Siberian Islands. Sediments discovered in ice cores confirm contact of sea ice with shallow waters in an early phase of growth. While in recent years the strong ice retreat in summer melts most of the shallow water ice on its way to the central Arctic Ocean (Krumpen et al., 2019), part of the residual ice encountered in the DNR has survived summer melt. Therefore, besides the original goals, MOSAiC will also provide an excellent opportunity to better understand the role of sea ice as a transport medium
for climate relevant gases, macro-nutrients, iron, organic matter, sediments, and pollutants from shelf areas to the central Arctic Ocean and beyond. This is particularly important because with predicted changes towards a seasonally ice-free ocean under climate change, a complete cut off of the long-range transport of ice-rafted materials by the Transpolar Drift appears possible in the future. By comparing transport rates of residual ice with newly formed ice on site, one can examine the impact a reduced long-range transport of sea ice has for the redistribution of biogeochemical matter in the central Arctic Ocean.

The application of the thermodynamic model reveals that ice in the DNR is 36% thinner than the surrounding ice due to its younger age and different provenance of origin. Differences in modal ice thickness between outer areas (sites R1-R3) and the DNR are also evident in direct field observations. It is therefore to be expected that the momentum and energy transfer between the ocean and the atmosphere is subject to strong spatial variations. Future studies will show whether these regional differences
can be reproduced using high-resolution models and satellite data. Whether the observed thickness gradients also influence ice dynamics in the immediate and extended surroundings of the Central Observatory is another exciting research question, and a comparison of the ice dynamics in the DNR and EMR derived from satellite data is work in progress. However, we assume that the encountered regional differences will balance out during the ice growth phase and thus reduce the spatial variability in ice dynamics over the course of the winter and over the course of the whole MOSAiC expedition.

The ice thickness in September 2019 can be classified as exceptionally thin when compared to the last 26 years. In this sense, we might have already experienced the "new normal" of Arctic conditions during the initial phase of MOSAiC, which might make future follow-up campaigns of this scale increasingly difficult. An only seasonally ice-covered Arctic with a reduced (or even cut off) transport of ice-rafted material by the Transpolar Drift will have strong implications for the redistribution of
biogeochemical matter in the central Arctic Ocean, with consequences for the balance of climate relevant trace gases, primary production, and biodiversity in the Arctic Ocean.

**Acknowledgement**

This work was carried out as part of the Russian-German Research Cooperation QUARCCS funded by the German Ministry for Education and Research (BMBF) under grant 03F0777A, and CATS under grant 63A0028B. Data used in this manuscript was produced as part of the international Multidisciplinary drifting Observatory for the Study of the Arctic Climate (MOSAiC) with the tag MOSAiC20192020 (AWI_PS122_1 and AF-MOSAiC-1_00). NCEP Reanalysis 2 data is made available by NOAA/OAR/ESRL PSD, Boulder, Colorado, USA, from their Web site at https://www.esrl.noaa.gov/psd/. The work on satellite remote sensing data was partly funded through the EU H2020 project SPICES (640161), the ESA Sea Ice CCI phase 1 and 2 (AO/1-6772/11/I-AM) and the Helmholtz PACES II (Polar regions And Coasts in the changing Earth System) and FRAM (FRontiers in Arctic marine Monitoring) program. TerraSAR-X images were provided by the German Aerospace Center (DLR) through TSX Science AO OCE3562. We thank the crew of the research vessel *Akademik Fedorov*, *Polarstern* and the helicopter company Naryan-Marsky for their great logistical support during the setup of the MOSAiC experiment and participants of the *Akademik Fedorov* cruise and MOSAiC School for helping hands.

**Code/data availability**

All data is archived in the MOSAiC Central Storage (MCS) and will be available on PANGAEA after finalization of the respective datasets according to the MOSAiC data policy. The gridded swath-processed CryoSat/SMOS datasets are available at https://www.meereisportal.de/. NCEP Reanalysis 2 data is made available by NOAA/OAR/ESRL PSD, Boulder, Colorado, USA, from their website at https://www.esrl.noaa.gov/psd/.

**Competing interests**

The authors declare no competing interests.

**Author contributions**

T.K. conceived the study and wrote the paper. F.B, F.K., S.H., J.B, V.S, L.v.A, C.H, T.R, R.R, V.S, E.D, A.T., J.H & S.S undertook the data analysis, developed the methods or contributed to interpretation of results. Field observations (thickness of snow and ice, bridge observations, ice cores, etc.) were made and processed by V.B, T.P, A.M., A.T, M.H, E.S, N.K, J.R, J.B, J.H, M.T & M.A. All authors commented on the manuscript.

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

| Site | Sampling device | Date | Ice thickness (m) | | | Snow thickness (m) | | | Total ice thickness (m) | |
|------|-----------------|------|------|------|---------|------|------|---------|------|------|
| | | | Mean | Mode | Samples | Mean | Mode | Samples | Mean | Mode |
| L1 | *GEM/Magna* | Oct. 05 | 0.86 (0.66) | 0.43 | 8.7 km | 0.10 (0.04) | 0.07 | n = 659 | 0.96 | 0.5 |
| L2 | *GEM/Magna* | Oct. 07 | 0.67 (0.54) | 0.33 | 9.6 km | 0.11 (0.04) | 0.08 | n = 519 | 0.78 | 0.41 |
| L3 | *GEM/Magna* | Oct. 09 | 1.0 (0.81) | 0.31 | 7.9 km | 0.11 (0.05) | 0.06 | n = 799 | 1.11 | 0.37 |
| M8 | *GEM/Magna* | Oct. 11 | 0.76 (0.75) | 0.35 | 1.2 km | 0.09 (0.04) | 0.06 | n = 385 | 0.85 | 0.41 |
| R1 | *GEM/Magna* | Oct. 01 | 0.85 (0.47) | 0.62 | 21 km | 0.11 (0.04) | 0.09 | n = 86 | 0.96 | 0.71 |
| R2 | *Manual* | Oct. 02 | 0.55 (0.1) | 0.60 | n = 38 | 0.18 | 0.18 | n = 38 | 0.73 | 0.78 |
| R3 | *Manual* | Oct. 02 | 0.61 (0.17) | 0.70 | n = 20 | 0.06 | 0.06 | n = 20 | 0.67 | 0.76 |


**Table 1:** Ice and snow thickness observations obtained on various residual ice floes in the immediate vicinity (grey, L1-L3, M8) and extended surroundings (R1-R3) of the Central Observatory. The positions of the sites are shown in Fig. 3a. Sample unit indicates either the distance covered by instruments like GEM/Magna (in km), or the number (n) of individual measurements that were performed manually. Numbers in parentheses provide the standard deviation.


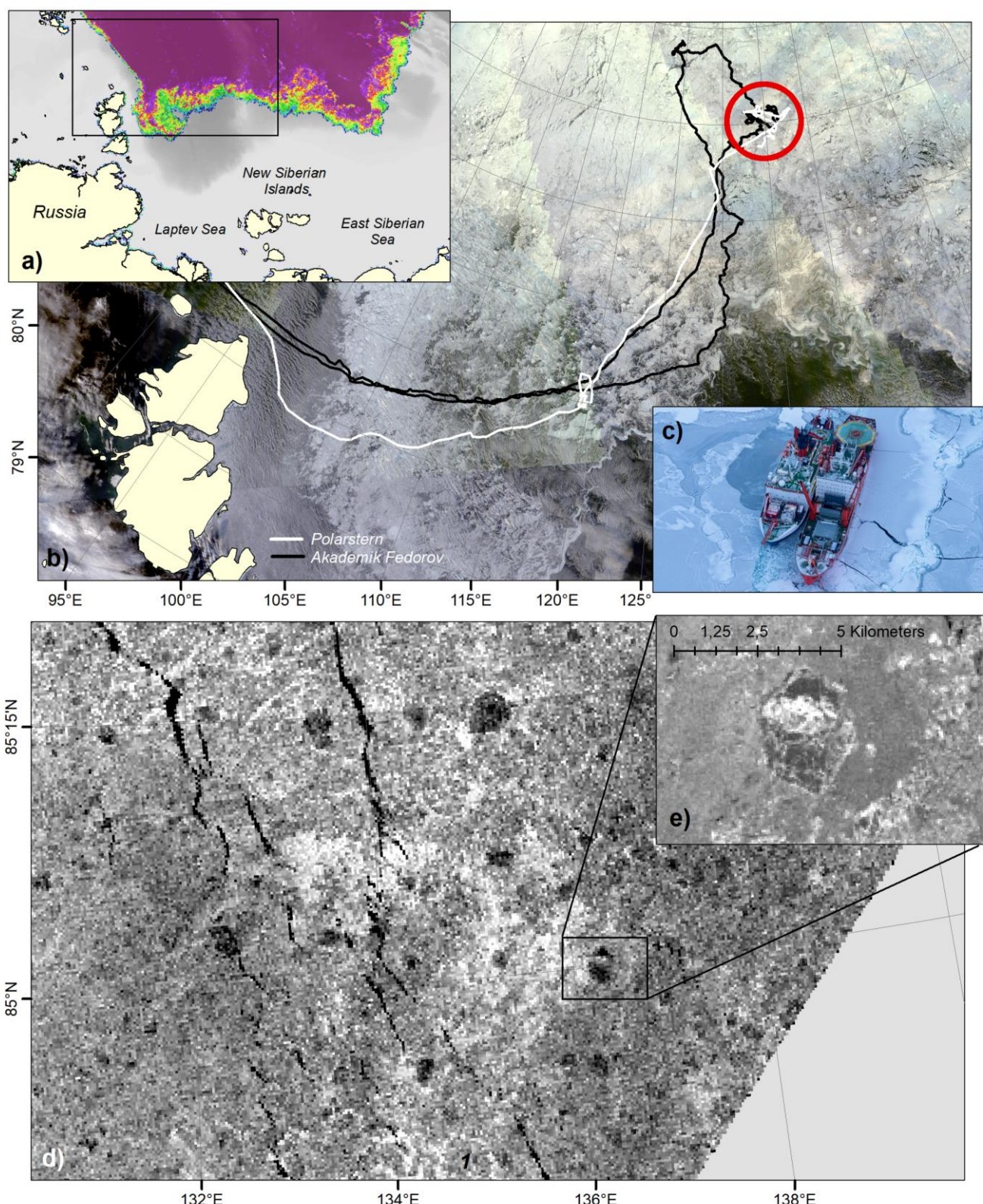

**Figure 1:** Initial sea ice conditions in the MOSAiC study region on the September 25, 2019, shortly before anchoring at the MOSAiC floe. a) Satellite-based sea ice concentration (source: University of Bremen). b) Ship tracks of *Polarstern* (white) and *Akademik Fedorov* (black) superimposed on a MODIS image (source: NASA) obtained on the 22 September 2019. The red circle indicates the Distributed Network Region (DNR, 40 km radius). c) *Akademik Fedorov* (right) and *Polarstern* (left) during bunkering procedure in thin ice, d) Sentinel-1 SAR image operated at C-band obtained on September 25 (source: ESA). The DN was mostly installed on the darker floes that correspond to older ice that had survived the summer (residual ice). The position of the Central Observatory is marked by a black rectangle. e) Close-up of the Central Observatory based on a TerraSAR-X image (X-band) obtained on September 25 (source: DLR). The floe was initially 2.8 x 3.8 km in size and is characterised by a strongly deformed zone in the centre, called the 'fortress'.


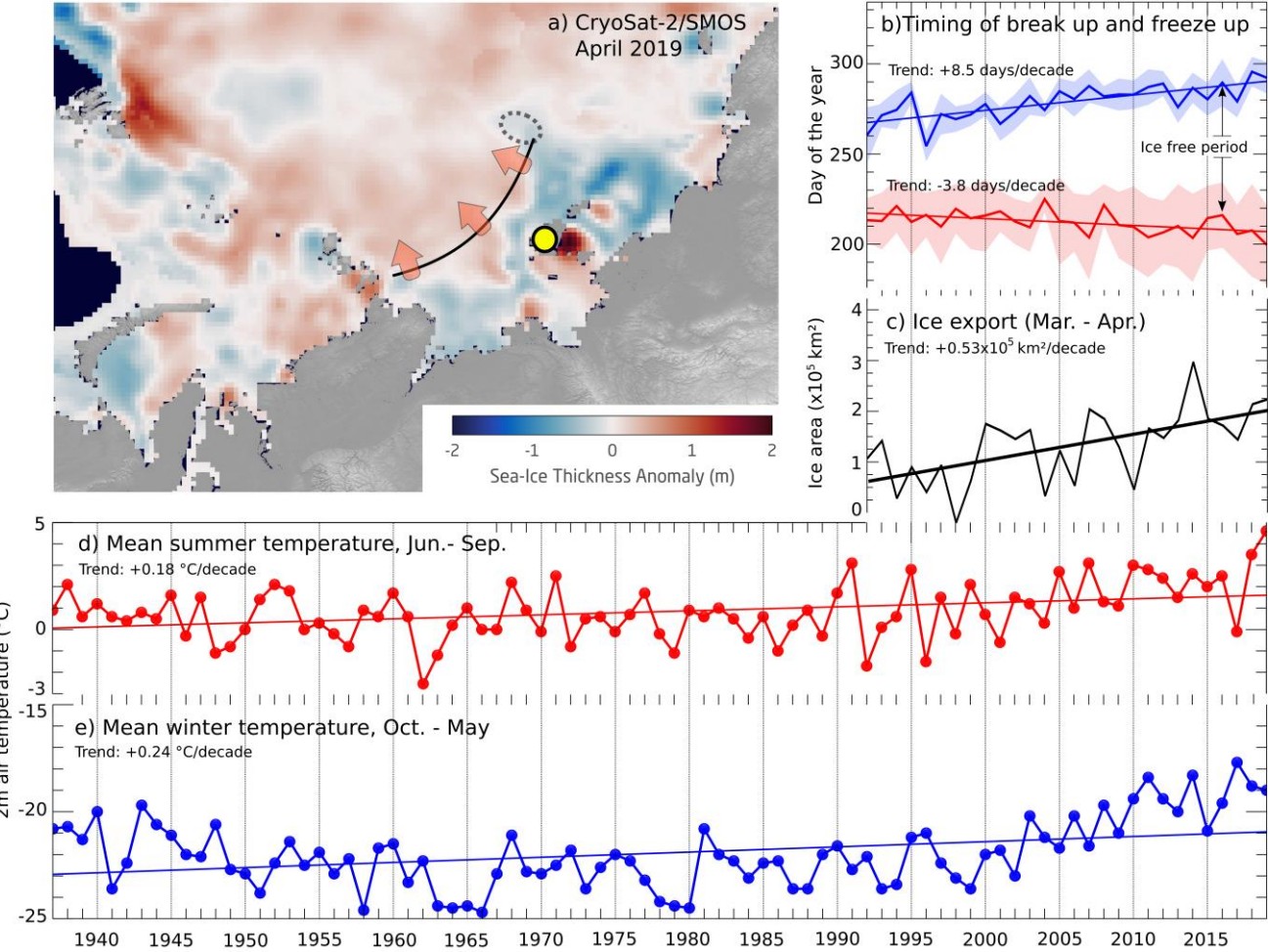

**Figure 2**: Summary of various processes that affected ice formation in the Laptev Sea and the East Siberian Sea in winter 2018/2019: a) CryoSat-2/SMOS sea ice thickness anomaly at the end of the winter (April 2019 minus April 2010 – 2018) in the eastern Eurasian Arctic. A zone of thinner ice was present prior to the onset of melting along the coastline. The ice field in which the MOSAiC expedition was set up 5 months later is marked by a dotted line. b) Estimate of the onset of break-up (red line) and freeze-up (blue line) with their standard deviations and trends between 86°N, 100°E and 71°N, 160°E. c) Satellite-based late winter (March – April) ice area export through a 'gate' spanning from 110°E to 160°E at 77.5°N. A trend line is plotted on top. In a), the gate is depicted as a solid black line. d,e) Air temperatures (2 m) recorded at Kotelny meteorological station (yellow circle in a) between 1935 and 2019 in the summer (red line) and winter months (blue line). All trends provided in this graph are significant at a 95% confidence level.

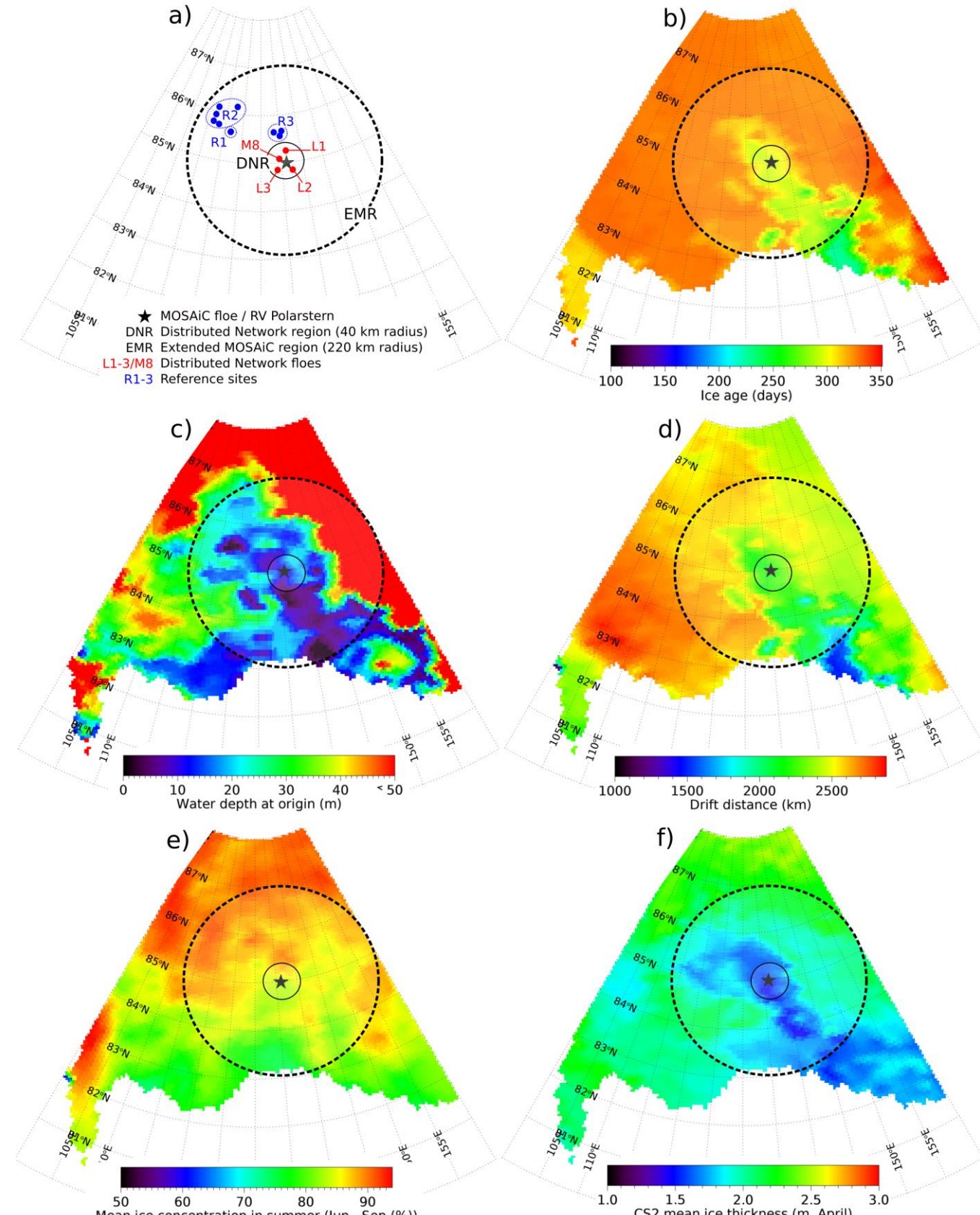

**Figure 3:** Results of Lagrangian sea ice backward tracking (see Methods). a) Starting point of the MOSAiC expedition (black star: position of the Central Observatory), the spatial extent of the investigation areas defined in this manuscript (DNR and EMR), and the Reference Sites where additional ice and snow thickness measurements were obtained. b) Sea ice age at the start of the MOSAiC expedition on September 25 according to Lagrangian tracking. c) Water depth at the ice formation site for each tracking position. d) Average distance of sea ice travelled from its formation site to its position on September 25. e) Sea ice concentration for each individual point, averaged over the first three months (June – September) of tracking along its trajectory. f) CryoSat-2 ice thickness estimates in late April, along the trajectory of each point.

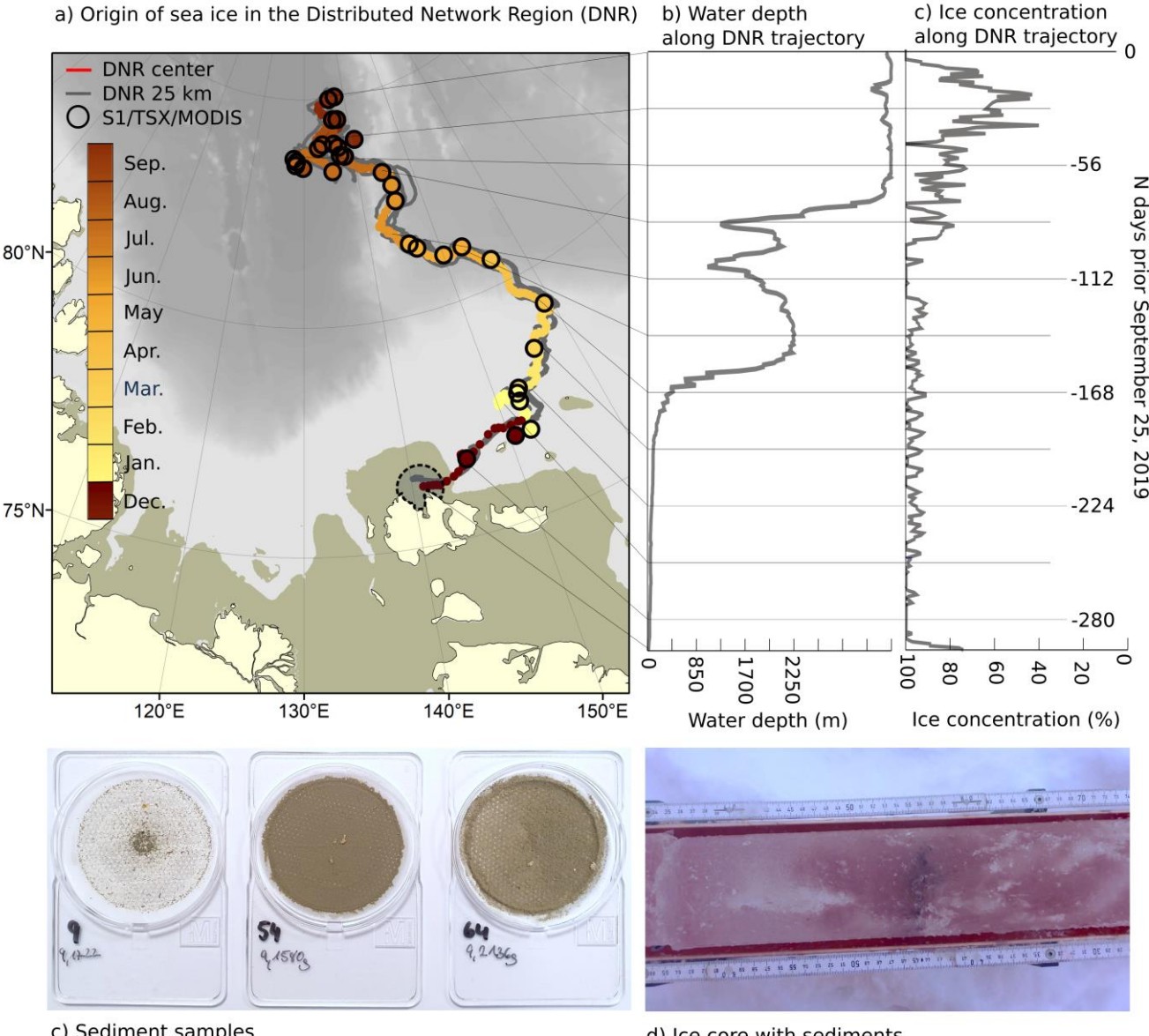

c) Sediment samples

d) Ice core with sediments

**Figure 4:** a) Lagrangian backward trajectories (see Methods) of the DNR. The multicolored trajectory line, with color corresponding to the month of year, indicates the center of the DNR (Central Observatory). The dashed circle provides the confidence bound of the ice origin. The grey lines provide additional trajectories for four points in the DNR at a distance of 25 km. Derived trajectories were verified by a manual tracking of the Central Observatory based on Sentinel-1, TerraSAR-X and MODIS (multicolored circles). The bathymetry is shown in the background. Brownish zones near the coast indicate shallow water areas of less then 30 m water depth. b) and c) show water depth (m) and ice concentration (%) along the trajectory of the Central Observatory. c) Sediment samples obtained from 10 cm ice core sections at L1 (left: level ice, 20-30 cm depth), L2 (middle: ridged/rafted ice, 243-253 cm depth, processed depth accounting for gaps in the core), and the central floe (right: ridged/rafted area at 49 – 59 cm depth). d) Ice core taken at the central floe (c, right) with a sediment layer.

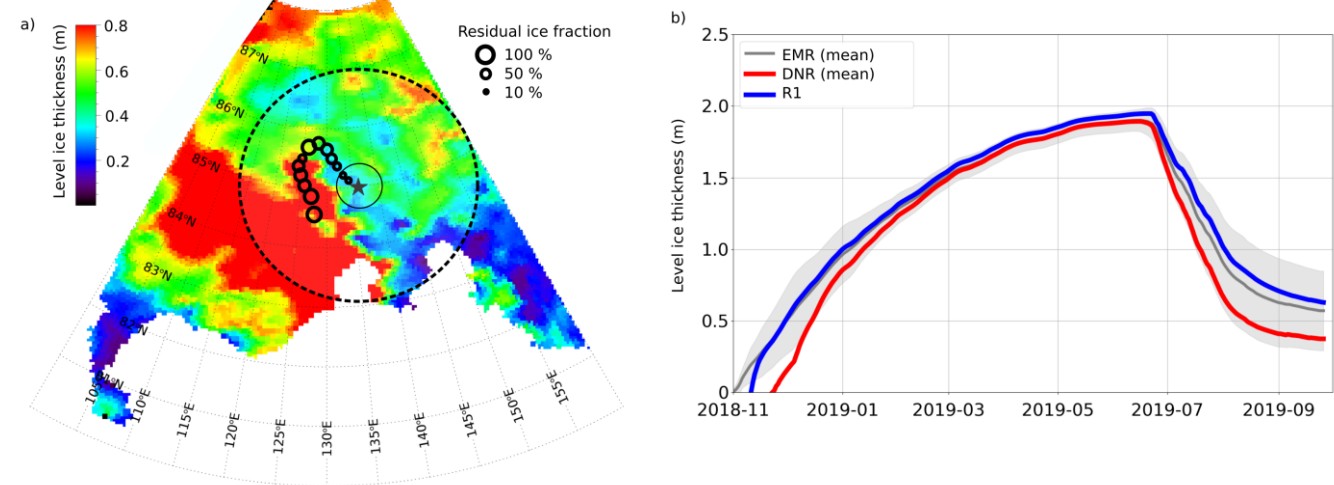

600

**Figure 5:** a) Level ice thickness on September 25, 2019 simulated with a thermodynamic model (see Methods). The percentage of residual ice observed along the course of *Akademik Fedorov* (black circles) is superimposed. b) Growth and melt of level ice in the EMR, DNR and at R1 (cf. Fig. 3a).

605

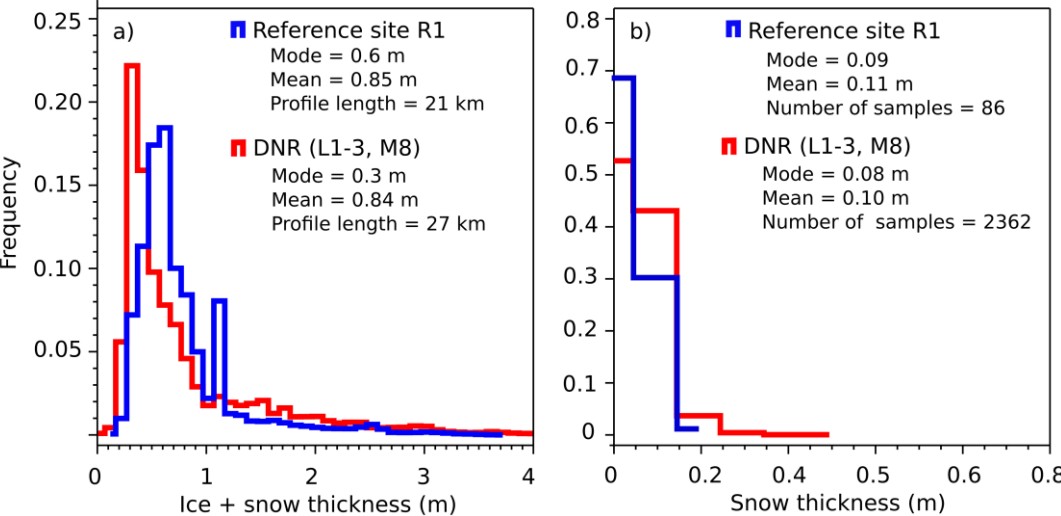

**Figure 6:** Total (ice plus snow, a) and snow (b) thickness distribution of the floes located inside the DNR (L1-3, M8, red line), and at R1 (blue line, see Fig. 3a for positions). Ice thickness measurements were made with a ground-based electromagnetic (GEM) instrument pulled across the ice on a sledge. Snow thickness measurements were made with a Magna Probe.

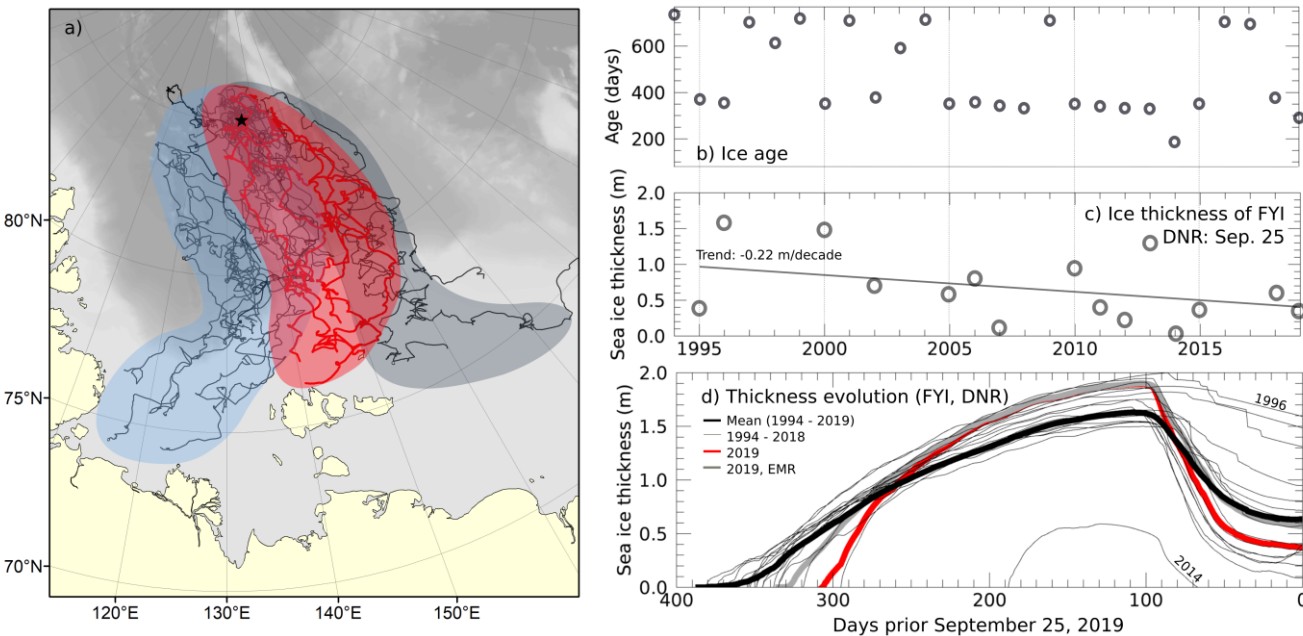

615 **Figure 7:** Ice origin, age and thickness of the DNR ice on September 25 between 1994 and 2019: a) Trajectories from the past 26 years separated by the region of origin: i) blue: Laptev Sea. ii) red: region north of the New Siberian Islands. iii) grey: East Siberian Sea. b) Age of the ice in the DNR region on September 25 of each year. c) Thickness of DNR FYI based on a thermodynamic model (Methods). d) Annual cycle of FYI growth and melt.