# Peer review of "The MOSAiC ice floe: sediment-laden survivor from the Siberian shelf"

_The Cryosphere, 2020_

## Referee Comment (RC1) · Anonymous Referee #1 · 21 Apr 2020

**The MOSAiC ice floe: sediment-laden survivor from the Siberian shelf**

by Thomas Krumpen and others

Submitted to The Cryosphere Discussions

**Review**

April 20, 2020

**Summary**

This manuscript presents a detailed analysis of the origins of the sea ice found within the neighborhood of the Polarstern at the beginning of the Multidisciplinary drifting Observatory for the Study of Arctic Climate (MOSAiC) drift experiment. It is the most

detailed provenance study for any region of sea ice that I know of and, as such, it is important both as a foundational paper for the vast body of sea ice work that will result from MOSAiC in the coming years and as an example of how provenance studies of drifting sea ice can be done. I congratulate the authors on preparing this manuscript so shortly after the MOSAiC floe was chosen. Overall the manuscript is concise and well written, but I have identified four areas where I feel the text is a little too brief and the reader would benefit from more detail. These are described in my major comments below and I believe they should all be easy to address.

**Major Comments**

1. Missing details for snow and ice thickness measurements

Section 2.5.1 omits important details regarding the measurement of snow and ice thickness. I understand that many of the details are described in the two references cited on lines 155-156, but as a foundational paper, I think it is important to provide the reader with more information. In particular, the text should state that the GEM method measures total snow-plus-ice thickness and the measured snow depth must be subtracted to derive the ice thickness. In addition, the text should describe the method used to align GEM and Magnaprobe measurements on a drifting floe and the resulting uncertainty in the ice thickness calculation

2. Missing details and uncertainty estimates for back trajectories

The estimated back trajectories for the ice in the DNR and EMR form the backbone of this manuscript. As a reader, I would therefore like to see more detailed describing the underlying methods and their associated uncertainties. I appreciate that the text includes citations for other studies that have used this method, but of the 5 citations listed on lines 99-100, only the final reference (Krumpen et al, 2016) contains any further detail. I recommend bringing this reference forward (i.e. replacing the reference on line 99) and including more of the relevant details in the text of this manuscript. In particular, I think it would be important to describe the uncertainties in the method, how

they vary during the season and how uncertainties are propagated backwards in time. It would also be helpful to illustrate this uncertainty in Figure 4, perhaps with an ellipse indicating the confidence bounds of the freeze-up region.

3. Ambiguity regarding use of high resolution satellite data to validate back trajctories

The close agreement between the back trajectory path and the black Xs in Figure 4a suggests that the uncertainties in the derived back trajectory of the MOSAiC ice are small. However, it is not clear from the text how the locations of the Xs were derived. The caption for Figure 4 states that the Xs represent locations determined from Sentinel-1, TerraSAR-X, and MODIS imagery. However, on lines 266-269, the text states that it was "extremely difficult to manually track the exact position of individual floes" and "only the approximate positions of individual floes could be determined". I recommend including further details regarding the nature of these difficulties, the manner in which approximate positions were determined, and the resulting uncertainties in ice position.

4. Location of Hass and Eicken data relative to study region

It is great to see that these early EM ice thickness data being used again for MOSAiC. However, I think it would be appropriate to either describe their location more precisely in the text (e.g. how close to the edge of the EMR) or show the location of these data on a map. For comparison with the GEM data collected here, it also be useful to state the total length of Haas and Eicken's measurement profiles.

**Minor comments**

Line 53 and all subsequent body text: Please add a vertical gap between each paragraph. This improves readability, particularly in longer sections with multiple paragraphs.

Line 100: Ricker et al (2018) is not included in the reference list

Line 145: Please clarify: does this mean that freeze-up and break-up are defined

according to when ice concentration exceeds and reaches zero, respectively?

Line 214: I suggest replacing "following" with "subsequent" to avoid leading any readers to think that the temperature anomalies will be described in the text that follows

Line 235: Assuming the authors are following standardized WMO nomenclature, I think it would be helpful to clarify that they are using "residual ice" as a shorthand for "residual first year ice", which does not graduate to become second year ice until January 1.

Line 248: I suggest replacing "closer nearby" with "closer to its location on September 25"

---

## Referee Comment (RC2) · Anonymous Referee #2 · 3 May 2020

Does the paper address relevant scientific questions within the scope of TC?: Yes Does the paper present novel concepts, ideas, tools, or data?: Yes Are substantial conclusions reached?: Yes Are the scientific methods and assumptions valid and clearly outlined?: Mostly yes, but some clarification needed in places. Are the results sufficient to support the interpretations and conclusions?: Yes Is the description of experiments and calculations sufficiently complete and precise to allow their reproduction by fellow scientists (traceability of results)? Do the authors give proper credit to related work and clearly indicate their own new/original contribution?: Yes Does the title clearly reflect the contents of the paper?: Yes Does the abstract provide a concise and complete summary?: Yes Is the overall presentation well structured and clear?: Yes Is the language fluent and precise?: Yes Are mathematical formulae, symbols, abbreviations,

and units correctly defined and used?: Yes Should any parts of the paper (text, formulae, figures, tables) be clarified, reduced, combined, or eliminated?: I've highlighted a few places where minor clarification would be helpful. Are the number and quality of references appropriate? Yes Is the amount and quality of supplementary material appropriate? Yes, though it might be interesting (if non-essential) to include some of the "not shown" work mentioned on line 314 here.

General comments: This is a well-written and very interesting manuscript exploring the origins of the ice floe that Polarstern was moored in for the MOSAiC expedition. Using a Lagrangian tracking technique in conjunction with a thermodynamic sea ice model, the authors establish the location of origin for the MOSAiC floe: North of the New Siberian Islands. The results described here will provide important context for future research based on MOSAiC. There are some places where the manuscript could be improved or clarified, but in general, these are minor. Occasionally, imprecise language is used – quantifying "a few" and "multiple" would be helpful, for instance. My main comment concerns the validation of the Lagrangian technique against satellite-derived floe positions. The computed trajectories appear to agree well with the satellite data (Figure 4a), but there is no measure of uncertainty in the satellite data included in this plot. Quantifying these uncertainties would be an important improvement. Additionally, neither the Lagrangian trajectories nor the satellite-derived positions have any indication of what time they were at each position in Figure 4a. Perhaps including a multicoloured trajectory line, with colour corresponding to day of year (and equivalent for the crosses) would aid the interpretation of this plot, and would give a better indication of how well the computed trajectories agree with the observations. Discussing how the discrepancy between computed trajectories and satellite observations evolves with time would provide important validation for the back-tracking results. Other places where I feel more explanation would strengthen the manuscript are detailed below in the specific comments section. I feel these comments are minor, and I look forward to seeing an updated version of the manuscript published in The Cryosphere.

Specific comments: Line 63: Have they really only been temporarily discontinued? With Arctic sea ice continuing to decline, surely the old style of drift stations have been permanently discontinued and replaced with new approaches like MOSAiC? Figure 1a: a box highlighting the region shown in Figure 1b would be helpful. Figure 1b: The term DNR is not mentioned until line 230 of the main text – as Figure 1 is discussed before this, better not to abbreviate in the caption, or to spell out what DNR stands for ~line 70. Line 76: Key citations(s) for the 'previously described methods' would be useful here. Line 82: Which atmospheric reanalysis data? Line 90: Which historical forcing data? Line 100: A sentence or two briefly summarising the findings of the cited papers would be nice to better put into context what IceTrack has previously been used for. Line 107: Is the 25km grid high enough resolution to capture smaller scale circulation features (e.g. eddies)? If not, how are the results likely to be impacted by not including these? Line 109: Why 40%, rather than the 20% threshold used in Krumpen et al. (2019)? Line 125: How many ice categories / layers? Table 1: I assume the numbers in parentheses after the means are a measure of the uncertainty, but it isn't clear from the caption whether this is standard deviation, range, or some other measure. Clarification needed. Line 187-190: I appreciate that that unavailability of oceanographic results means that it is necessary to focus on atmospherically driven processes affecting the retreat of the ice edge in 2019. However, a short overview of the oceanic processes that play an important role in general (rather than specifically 2019) would help to better put this work into context. Line 204: In Figure 2a, there is a positive thickness anomaly immediately around the New Siberian Islands. The trend for the region in general is clearly negative, but 'negative thickness anomalies throughout the entire coastal zones' is phrased too strongly – the exception to the trend should at least be noted. Line 245: How many is "a few" days? It would be great if an estimate is possible, if not perhaps better to drop. Line 269: Given that only approximate locations of floes could be determined from the high (how high?) resolution satellite data, it would be good to see some measure of the uncertainty attached to the crosses shown in Figure 4a, or at least discussed in the text. Figure 4: Linked to my previous comment, it would also

be helpful to quantify how closely the satellite-derived positions match the computed trajectories. It is already clear from Figure 4a that they follow similar paths, but this doesn't necessarily show that they're in (approximately) the same place at the same time. Including a measure of the distance between the Xs and the corresponding point on the Lagrangian trajectories would answer this question, and would be interesting to compare to the (36 +/-20 km after 200 days) deviation found in previous work. Line 314: It would be interesting to include a brief overview of the 'not shown' analysis in the supplementary material. Line 318: How close was the ARK-12 cruise data "in the surroundings of the DNR" to the DNR?

Technical corrections: Line 187: I think a word is missing (e.g. "In the following section, . . .") Line 192: This reads a bit strangely. Perhaps "Ice dynamics and ice export in winter are important preconditioning mechanisms for the ice retreat in summer" would be better.

---

## Author Comment (AC1) · 18 May 2020

Dear Reviewer 1,

thank you very much for your helpful comments and suggestions. Your feedback is very much appreciated.

We agree with the fact that the method description is sometimes a little short. This has also been noted by the second reviewer as well. We will go into more detail at the points you mentioned.

With respect to your major comments made:

1. Missing details for snow and ice thickness measurements

We agree. The revised manuscript will better describe the methodology (+ uncertainties) to derive thickness from GEM and how surveys were aligned with snow samples obtained along track.

2. Missing details and uncertainty estimates for back trajectories

In the revised manuscript, we will include a discussion about uncertainties of the various products, both, in summer and in winter. Moreover, we will include a comparison of real buoys with IceTrack trajectories to outline the performance of the Lagrangian approach and number uncertainties related to the identification of the source areas.

3. Ambiguity regarding use of high resolution satellite data to validate back trajectories

This aspect will be better described in the revised manuscript. Since the MOSAiC floe itself was difficult to identify on single images, large scale patterns (ridge systems etc) of several tens of kilometers were tracked. The description will be revised. Thanks for the hint. Together with a better description of the uncertainties of the applied motion data (comment 2), this should give the reader a better understanding of the limitations of the applied methods

4. Location of Hass and Eicken data relative to study region

We will either better describe location or, if possible, include positions in one of the maps. We agree that profile length of surveys carried out by Haas and Eicken must be provided.

5. With respect to all other comments:

Thank you very much. We will take all minor comments into account.

Once again many thanks for the careful revision of the manuscript and valuable feedback.

With best regards on behalf of all co-authors Thomas Krumpen

---

## Author Comment (AC2) · 18 May 2020

Dear Reviewer 2,

thank you very much for your helpful comments and suggestions. Your feedback is very much appreciated.

We agree with the fact that the method description is sometimes a little short. This has also been noted by the second reviewer as well. We will go into more detail at the points you mentioned.

With respect to your general comments made:

1. Validation of the Lagrangian technique

[Figure]

These concerns were also addressed by Reviewer 1. We agree that the revised manuscript should include a discussion about uncertainties of the various motion products, both, in summer and in winter. Moreover, we will include a comparison of real buoys with IceTrack trajectories to outline the performance of the Lagrangian approach and number uncertainties related to the identification of the source areas. Furthermore, we will better describe the visual identification of features on high resolution data used to validate back trajectories. The trajectories showing the origin of the ice floe will also include a color scale showing where the ice was at what time.

With respect to your specific comments made:

We will address all minor comments made. Below a brief reply to some of your specific comments which are more critical:

1. Line 109: Why 40%, rather than 20% threshold:

The idea of using a higher threshold was to better represent the influence of areas of ice production like polynyas on the trajectories. However, the trajectories do not differ, no matter if 40% or 20% is used.

2. Oceanic processes affecting sea ice retreat in summer:

We will add a short discussion about the impact of the ocean on sea ice retreat.

3. Closeness of the ARK-12 cruise data: We will either better describe location or, if possible, include positions in one of the maps.

Thank you very much. We will take all minor comments into account. Imprecise Language will be revised where it's necessary ("few", "multiple", etc)

Once again many thanks for the careful revision of the manuscript and valuable feedback.

With best regards on behalf of all co-authors Thomas Krumpen

---

## Author Response (AR1)

Title: The MOSAiC ice floe: sediment-laden survivor from the Siberian shelf
Author(s): Thomas Krumpen et al.
MS No.: tc-2020-64
MS Type: Research article
Iteration: Minor Revision

Dear Reviewer 1,

thank you very much for your helpful comments and suggestions. Your feedback is very much appreciated. We agree with the fact that the method description is sometimes a little short. This has also been noted by the second reviewer as well. We will go into more detail at the points you mentioned.

With respect to your major comments made:

1.       *Missing details for snow and ice thickness measurements: Section 2.5.1 omits important details regarding the measurement of snow and ice thickness. I understand that many of the details are described in the two references cited on lines 155-156, but as a foundational paper, I think it is important to provide the reader with more information. In particular, the text should state that the GEM method measures total snow-plus-ice thickness and the measured snow depth must be subtracted to derive the ice thickness. In addition, the text should describe the method used to align GEM and Magnaprobe measurements on a drifting floe and the resulting uncertainty in the ice thickness calculation.*

We agree. We improved the description of the methodology (section 2.5.1) clearly stating that the GEM measures total snow-plus-ice thickness. In addition we refers to the study by Haas and Eicken (2001) who compared drill hole measurements with GEM measurements to number uncertainties of electromagnetic sounding during summer month in the central Arctic Ocean. Moreover we state that both, GEM and Magna Probe measurements were projected into a common drift and rotation corrected coordinate system (using data from a GPS base station), before we could do the alignment.

2.       *Missing details and uncertainty estimates for back trajectories The estimated back trajectories for the ice in the DNR and EMR form the backbone of this manuscript. As a reader, I would therefore like to see more detailed describing the underlying methods and their associated uncertainties. I appreciate that the text includes citations for other studies that have used this method, but of the 5 citations listed on lines 99-100, only the final reference (Krumpen et al, 2016) contains any further detail. I recommend bringing this reference forward (i.e. replacing the reference on line 99) and including more of the relevant details in the text of this manuscript. In particular, I think it would be important to describe the uncertainties in the method, how they vary during the season and how uncertainties are propagated backwards in time. It would also be helpful to illustrate this uncertainty in Figure 4, perhaps with an ellipse indicating the confidence bounds of the freeze-up region.*

We agree. Since the trajectories are an important element of the manuscript, associated uncertainties need to be discussed or even evaluated through a comparison with in-situ data. In the revised manuscript, we included a more detailed discussion about uncertainties of the low resolution drift products (see changes made in chapter 2.1 and 3.2). Actually we go one step further: In order to provide more accurate uncertainty estimates for the backtracking and source area deamination, we

reproduce the drift of real GPS buoys with IceTrack. This has been done earlier in Krumpen et al. (2019) using position data from ~60 buoys deployed in the central Arctic Ocean. According to the authors, the deviation between actual and virtual tracks is rather small (36 km +/- 0 km after 200 days), but this "forward-directed" comparison is mainly based on buoys deployed at the end of the summer. However, in this study we perform a "backward" tracking starting at the end of the summer near the ice edge. Hence, uncertainties of IceTrack trajectories may be higher, since motion products are less accurate during summer month and near the ice edge. We have therefore repeated the validation, taking into account buoys that have survived a complete summer and winter (there are not many of them available, but we identified 10). The drift of these buoys was reproduced with IceTrack (see new Fig. C which was added to the appendix). The figure indeed shows higher uncertainties than in Krumpen et al (2019): 60 +/-24 km after 320 days. Note that the maximum deviation between real and virtual buoys gives a measure of the largest possible error that can occur when determining the ice origin. After 320 days it is around 105 km. Following your suggestion, the confidence bound is now shown in Fig. 4 as an ellipsoid (dashed line).

[Figure]

**New Figure C in Supplement**: Results from backward tracking of buoy locations: Distance between 10 buoys (source: seaiceportal.de) deployed on sea ice in the Arctic Ocean between 2015 and 2018 and their reconstructed trajectories (virtual buoys).

[Figure]

**New Figure 4 with confidence bound**: a) Lagrangian backward trajectories (see Methods) of the DNR. The multicolored trajectory line, with color corresponding to the month of year, indicates the center of the DNR (Central Observatory). The dashed circle provides the confidence bound of the ice origin. The grey lines provide additional trajectories for four points in the DNR at a distance of 25 km. Derived trajectories were verified by a manual tracking of the Central Observatory based on Sentinel-1, TerraSAR-X and MODIS (multicolored circles). The bathymetry is shown in the background. Brownish zones near the coast indicate shallow water areas of less then 30 m water depth. b) and c) show water depth (m) and ice

concentration (%) along the trajectory of the Central Observatory. c) Sediment samples obtained from 10 cm ice core sections at L1 (left: level ice, 20-30 cm depth), L2 (middle: ridged/rafted ice, 243-253 cm depth, processed depth accounting for gaps in the core), and the central floe (right: ridged/rafted area at 49 – 59 cm depth). d) Ice core taken at the central floe (c, right) with a sediment layer.

*3.      Ambiguity regarding use of high resolution satellite data to validate back trajectories: The close agreement between the back trajectory path and the black Xs in Figure 4a suggests that the uncertainties in the derived back trajectory of the MOSAiC ice are small. However, it is not clear from the text how the locations of the Xs were derived. The caption for Figure 4 states that the Xs represent locations determined from Sentinel-1, TerraSAR-X, and MODIS imagery. However, on lines 266-269, the text states that it was "extremely difficult to manually track the exact position of individual floes" and "only the approximate positions of individual floes could be determined". I recommend including further details regarding the nature of these difficulties, the manner in which approximate positions were determined, and the resulting uncertainties in ice position.*

Thanks for pointing this out. Although it was not possible to track the exact position of the floe, it was still possible to trace textures of larger ice structures such as shear zones. This aspect is now better described in the revised manuscript. Together with a better description and assessment of the uncertainties of the applied motion data (comment 2), this should give the reader a better understanding of the limitations of the applied methods.

*4.      Location of Hass and Eicken data relative to study region: It is great to see that these early EM ice thickness data being used again for MOSAiC. However, I think it would be appropriate to either describe their location more precisely in the text (e.g. how close to the edge of the EMR) or show the location of these data on a map. For comparison with the GEM data collected here, it also be useful to state the total length of Haas and Eicken's measurement profiles.*

We better described the location and profile length. Note that results obtained by Haas and Eicken (2001) are now introduced in the beginning of chapter 3.3, since their findings nicely support the notion of exceptionally thin ice in the MOSAiC starting region. When revising the paragraph with reference to Haas and Eicken, we also noticed an incorrectly stated value. The observed modal thickness was 1.85 m in the EMR, not 2.1. This further improves agreement with our thermodynamic model (1.6 m).

*5.      Response to minor comments:*

*Line 53 and all subsequent body text: Please add a vertical gap between each paragraph. This improves readability, particularly in longer sections with multiple paragraphs.*

Done

*Line 100: Ricker et al (2018) is not included in the reference list*

Thanks, added.

*Line 145: Please clarify: does this mean that freeze-up and break-up are defined according to when ice concentration exceeds and reaches zero, respectively?*

Yes. We better clarify definition in the text. Thanks for pointing this out.

*Line 214: I suggest replacing "following" with "subsequent" to avoid leading any readers to think that the temperature anomalies will be described in the text that follows*

Thanks, corrected.

*Line 235: Assuming the authors are following standardized WMO nomenclature, I think it would be helpful to clarify that they are using "residual ice" as a shorthand for "residual first year ice", which does not graduate to become second year ice until January 1.*

That is correct. We have included your comment in the text.

*Line 248: I suggest replacing "closer nearby" with "closer to its location on September 25ᵗʰ.*

We agree. Corrected

Once again many thanks for the careful revision of the manuscript and valuable feedback.

With best regards

on behalf of all co-authors

Thomas Krumpen

Title: The MOSAiC ice floe: sediment-laden survivor from the Siberian shelf
Author(s): Thomas Krumpen et al.
MS No.: tc-2020-64
MS Type: Research article
Iteration: Minor Revision

Dear Reviewer 2,

thank you very much for your helpful comments and suggestions. Your feedback is very much appreciated.

With respect to your general comments made:

*My main comment concerns the validation of the Lagrangian technique against satellite-derived floe positions. The computed trajectories appear to agree well with the satellite data (Figure 4a), but there is no measure of uncertainty in the satellite data included in this plot. Quantifying these uncertainties would be an important improvement. Additionally, neither the Lagrangian trajectories nor the satellite-derived positions have any indication of what time they were at each position in Figure 4a. Perhaps including a multicolored trajectory line, with color corresponding to day of year (and equivalent for the crosses) would aid the interpretation of this plot, and would give a better indication of how well the computed trajectories agree with the observations. Discussing how the discrepancy between computed trajectories and satellite observations evolves with time would provide important validation for the back-tracking results.*

We agree. Same concerns were brought up by Reviewer 1. Since the trajectories are an important element of the manuscript, associated uncertainties need to be discussed or even evaluated through a comparison with in-situ data. In the revised manuscript, we included a more detailed discussion about uncertainties of the low resolution drift products (see changes made in chapter 2.1 and 3.2). Actually we go one step further: In order to provide more accurate uncertainty estimates for the backtracking and source area deamination, we reproduce the drift of real GPS buoys with IceTrack. This has been done earlier in Krumpen et al. (2019) using position data from ~60 buoys deployed in the central Arctic Ocean. According to the authors, the deviation between actual and virtual tracks is rather small (36 km +/- 0 km after 200 days), but this "forward-directed" comparison is mainly based on buoys deployed at the end of the summer. However, in this study we perform a "backward" tracking starting at the end of the summer near the ice edge. Hence, uncertainties of IceTrack trajectories may be higher, since motion products are less accurate during summer month and near the ice edge. We have therefore repeated the validation, taking into account buoys that have survived a complete summer and winter (there are not many of them available, but we identified 10). The drift of these buoys was reproduced with IceTrack (see new Fig. C which was added to the appendix). The figure indeed shows higher uncertainties than in Krumpen et al (2019): 60 +/-24 km after 320 days. Note that the maximum deviation between real and virtual buoys gives a measure of the largest possible error that can occur when determining the ice origin. After 320 days it is around 105 km. The "confidence bound" is now shown in Fig. 4 as an ellipsoid (dashed line).

Following your suggestion, we also included a multicolored trajectory line with color corresponding to the month of year (and equivalent for position estimates based on high resolution satellite data; note that crosses turned into circles).

[Figure]

**New Figure C in Supplement**: Results from backward tracking of buoy locations: Distance between 10 buoys (source: seaiceportal.de) deployed on sea ice in the Arctic Ocean between 2015 and 2018 and their reconstructed trajectories (virtual buoys).

[Figure]

**New Figure 4 with confidence bound**: a) Lagrangian backward trajectories (see Methods) of the DNR. The multicolored trajectory line, with color corresponding to the month of year, indicates the center of the DNR (Central Observatory). The dashed circle provides the confidence bound of the ice origin. The grey lines provide additional trajectories for four points in the DNR at a distance of 25 km. Derived trajectories were verified by a manual tracking of the Central Observatory based on Sentinel-1, TerraSAR-X and MODIS (multicolored circles). The bathymetry is shown in the background. Brownish zones near the coast indicate shallow water areas of less then 30 m water depth. b) and c) show water depth (m) and ice concentration (%) along the trajectory of the Central Observatory. c) Sediment samples obtained from 10 cm ice core sections at L1 (left: level ice, 20-30 cm depth),

With respect to your specific comments made:

*Line 63: Have they really only been temporarily discontinued? With Arctic sea ice continuing to decline, surely the old style of drift stations have been permanently discontinued and replaced with new approaches like MOSAiC?*

Yes. Russia has just started the construction of the "North Pole" station. The platform will be permanently drifting in high Arctic waters.

*Figure 1a: a box highlighting the region shown in Figure 1b would be helpful.*

Thanks. We agree, this is helpful. A box was added to Fig. 1a.

*Figure 1b: The term DNR is not mentioned until line 230 of the main text – as Figure 1 is discussed before this, better not to abbreviate in the caption, or to spell out what DNR stands for ~line70.*

We now explain the acronym DNR in the caption of Fig. 1.

*Line 76: Key citations(s) for the 'previously described methods' would be useful here.*

Good point. Key citations were added to the introduction.

*Line 82: Which atmospheric reanalysis data?*

Its NCEP (added).

*Line 90: Which historical forcing data?*

Its NCEP that is missing here. Thanks for the hint. "!Historical" was replaced by "NCEP".

*Line 100: A sentence or two briefly summarizing the findings of the cited papers would be nice to better put into context what IceTrack has previously been used for.*

The cited papers used IceTrack for the same purpose as we do in this study: To determine the origin, pathways and thickness changes of sea ice, as well as the atmospheric forcing acting on the ice cover. We make this more clear now.

*Line 107: Is the 25km grid high enough resolution to capture smaller scale circulation features (e.g. eddies)? If not, how are the results likely to be impacted by not including these?*

This is an interesting question, but difficult to answer here. Actually the grid resolution varies with the applied motion product (25 km = NSIDC Pathfinder vs. 62.5 km OSISAF), and as such most likely the ability to resolve small scale features. It would possibly require a comparison of IceTrack derived trajectories with results from some high resolution model (work in progress by NOC).

*Line 109: Why 40%, rather than the 20% threshold used in Krumpen et al.(2019)?*

The algorithm stops tracking procedure (in backward mode), if shallow water is reached or if ice concentration at a specific location drops below the above-mentioned thresholds. In theory, this indicates that the parcel has entered an area where ice is actively produced. However, ice concentration in polynyas are usually a bit higher than 20 %, which is why we were testing a different threshold. Nevertheless, no significant differences in sea ice pathways and source areas were observed when repeating the using higher/lower ice concentration thresholds or different combinations of motion products. This is now clearly stated in chapter 3.2 (after discussing the uncertainties of IceTrack estimates).

*Line 125: How many ice categories / layers?*

Seven categories and seven layers. We added this information to the manuscript.

*Table 1: I assume the numbers in parentheses after the means are a measure of the uncertainty, but it isn't clear from the caption whether this is standard deviation, range, or some other measure. Clarification needed.*

Yes, it's the standard deviation. We provide clarification in Tab. 1, caption.

*Line 187-190: I appreciate that that unavailability of oceanographic results means that it is necessary to focus on atmospherically driven processes affecting the retreat of the ice edge in 2019. However, a short overview of the oceanic processes that play an important role in general (rather than specifically 2019) would help to better put this work into context.*

We now also refer to oceanic processes and preconditioning mechanisms (including references) that play an important role at that time of the year and in this area:

- ….enhanced winter ventilation of the ocean that can reduce sea ice formation in this area at a rate now comparable to losses from atmospheric thermodynamic forcing (Polyakov et al., 2017).
- …the interaction between surface winds and warm sea surface temperatures in areas from which the ice has already retreated (Steele and Ermold, 2015).
- …intensive warming of the upper ocean that causes a delay in the autumnal freeze-up of sea ice (e.g. Janout et al., 2016)

*Line 204: In Figure 2a, there is a positive thickness anomaly immediately around the New Siberian Islands. The trend for the region in general is clearly negative, but 'negative thickness anomalies throughout the entire coastal zones' is phrased too strongly – the exception to the trend should at least be noted.*

We added "except for southern half of the area around the New Siberian Islands" to the sentence. This should make the statement less strong. Note that we were discussing the reason for the positive anomaly south of the New Siberian Islands, which is dominated by fast ice at that time of the year. It can be either an artificial feature caused by processing (e.g. along-track sea surface height interpolation errors), or a real thick ice fraction as a consequence of intensified ice dynamics.

*Line 245: How many is "a few" days? It would be great if an estimate is possible, if not perhaps better to drop.*

It's difficult to date the onset exactly. Hence, we decided to drop it.

*Line 269: Given that only approximate locations of floes could be determined from the high (how high?) resolution satellite data, it would be good to see some measure of the uncertainty attached to the crosses shown in Figure 4a, or at least discussed in the text. AND Figure 4: Linked to my previous comment, it would also be helpful to quantify how closely the satellite-derived positions match the computed trajectories. It is already clear from Figure 4a that they follow similar paths, but this doesn't necessarily show that they're in (approximately) the same place at the same time. Including a measure of the distance between the Xs and the corresponding point on the Lagrangian trajectories would answer this question, and would be interesting to compare to the (36 +/-20 km after 200 days) deviation found in previous work.*

We decided to put less focus on a discussion of high resolution data itself, since position estimates are approximations only and difficult to cross-compare. Instead, we introduced a separate validation of lagrangian trajectories (IceTrack).  Please see our reply to your general comment and the 3[rd] general comment of reviewer 1. However, not that we changed Fig 4 following your suggestions and added a color coding to position estimates based on high resolution satellite data. Through this, it should become more clear, that trajectories based on MODIS/TSX/S1 are very similar to estimates obtained from IceTrack.

*Line 314: It would be interesting to include a brief overview of the 'not shown' analysis in the supplementary material.*

Yes, fully agree. Actually, the thermal and solar radiation anomaly is less striking. What is however quite remarkable are the very low precipitation rates ($2^{nd}$ lowest value compared to other years), leading to intensified ice growth. We added a figure showing the accumulated precipitation rates along different trajectories to the Appendix (Fig. D).

[Figure]

**Figure D, Supplement**: Snow accumulation along trajectories between 1994 – 2019 (FYI only, compare Fig. 7). Precipitation rates are obtained from NCEP reanalysis data.

*Line 318: How close was the ARK-12 cruise data "in the surroundings of the DNR" to the DNR?*

Please see our answer to the $4^{th}$ general comment of reviewer one. We now provide a better description of the ARK-12 data, including a more precise definition of the working area and profile length.

*With respect to your technical comments made*:

*Line 187: I think a word is missing (e.g. "In the following section,...")*

Thanks. Corrected

*Line 192: This reads a bit strangely. Perhaps "Ice dynamics and ice export inwinter are important preconditioning mechanisms for the ice retreat in summer" would be better.*

Fully agree. We use your formulation in the revised manuscript.

Once again many thanks for the careful revision of the manuscript and valuable feedback.

With best regards

on behalf of all co-authors

Thomas Krumpen

---

## Editor Decision (ED1)

**The MOSAiC ice floe: sediment-laden survivor from the Siberian shelf**

by Thomas Krumpen and others

Submitted to The Cryosphere Discussions

**Review**

April 20, 2020

**Summary**

This manuscript presents a detailed analysis of the origins of the sea ice found within the neighborhood of the Polarstern at the beginning of the Multidisciplinary drifting Observatory for the Study of Arctic Climate (MOSAiC) drift experiment. It is the most

detailed provenance study for any region of sea ice that I know of and, as such, it is important both as a foundational paper for the vast body of sea ice work that will result from MOSAiC in the coming years and as an example of how provenance studies of drifting sea ice can be done. I congratulate the authors on preparing this manuscript so shortly after the MOSAiC floe was chosen. Overall the manuscript is concise and well written, but I have identified four areas where I feel the text is a little too brief and the reader would benefit from more detail. These are described in my major comments below and I believe they should all be easy to address.

**Major Comments**

1. Missing details for snow and ice thickness measurements

Section 2.5.1 omits important details regarding the measurement of snow and ice thickness. I understand that many of the details are described in the two references cited on lines 155-156, but as a foundational paper, I think it is important to provide the reader with more information. In particular, the text should state that the GEM method measures total snow-plus-ice thickness and the measured snow depth must be subtracted to derive the ice thickness. In addition, the text should describe the method used to align GEM and Magnaprobe measurements on a drifting floe and the resulting uncertainty in the ice thickness calculation

2. Missing details and uncertainty estimates for back trajectories

The estimated back trajectories for the ice in the DNR and EMR form the backbone of this manuscript. As a reader, I would therefore like to see more detailed describing the underlying methods and their associated uncertainties. I appreciate that the text includes citations for other studies that have used this method, but of the 5 citations listed on lines 99-100, only the final reference (Krumpen et al, 2016) contains any further detail. I recommend bringing this reference forward (i.e. replacing the reference on line 99) and including more of the relevant details in the text of this manuscript. In particular, I think it would be important to describe the uncertainties in the method, how

they vary during the season and how uncertainties are propagated backwards in time. It would also be helpful to illustrate this uncertainty in Figure 4, perhaps with an ellipse indicating the confidence bounds of the freeze-up region.

**3. Ambiguity regarding use of high resolution satellite data to validate back trajctories**

The close agreement between the back trajectory path and the black Xs in Figure 4a suggests that the uncertainties in the derived back trajectory of the MOSAiC ice are small. However, it is not clear from the text how the locations of the Xs were derived. The caption for Figure 4 states that the Xs represent locations determined from Sentinel-1, TerraSAR-X, and MODIS imagery. However, on lines 266-269, the text states that it was "extremely difficult to manually track the exact position of individual floes" and "only the approximate positions of individual floes could be determined". I recommend including further details regarding the nature of these difficulties, the manner in which approximate positions were determined, and the resulting uncertainties in ice position.

**4. Location of Hass and Eicken data relative to study region**

It is great to see that these early EM ice thickness data being used again for MOSAiC. However, I think it would be appropriate to either describe their location more precisely in the text (e.g. how close to the edge of the EMR) or show the location of these data on a map. For comparison with the GEM data collected here, it also be useful to state the total length of Haas and Eicken's measurement profiles.

**Minor comments**

Line 53 and all subsequent body text: Please add a vertical gap between each paragraph. This improves readability, particularly in longer sections with multiple paragraphs.

Line 100: Ricker et al (2018) is not included in the reference list

Line 145: Please clarify: does this mean that freeze-up and break-up are defined

according to when ice concentration exceeds and reaches zero, respectively?

Line 214: I suggest replacing "following" with "subsequent" to avoid leading any readers to think that the temperature anomalies will be described in the text that follows

Line 235: Assuming the authors are following standardized WMO nomenclature, I think it would be helpful to clarify that they are using "residual ice" as a shorthand for "residual first year ice", which does not graduate to become second year ice until January 1.

Line 248: I suggest replacing "closer nearby" with "closer to its location on September 25"

[Figure]

Does the paper address relevant scientific questions within the scope of TC?: Yes Does the paper present novel concepts, ideas, tools, or data?: Yes Are substantial conclusions reached?: Yes Are the scientific methods and assumptions valid and clearly outlined?: Mostly yes, but some clarification needed in places. Are the results sufficient to support the interpretations and conclusions?: Yes Is the description of experiments and calculations sufficiently complete and precise to allow their reproduction by fellow scientists (traceability of results)? Do the authors give proper credit to related work and clearly indicate their own new/original contribution?: Yes Does the title clearly reflect the contents of the paper?: Yes Does the abstract provide a concise and complete summary?: Yes Is the overall presentation well structured and clear?: Yes Is the language fluent and precise?: Yes Are mathematical formulae, symbols, abbreviations,

and units correctly defined and used?: Yes Should any parts of the paper (text, formulae, figures, tables) be clarified, reduced, combined, or eliminated?: I've highlighted a few places where minor clarification would be helpful. Are the number and quality of references appropriate? Yes Is the amount and quality of supplementary material appropriate? Yes, though it might be interesting (if non-essential) to include some of the "not shown" work mentioned on line 314 here.

General comments: This is a well-written and very interesting manuscript exploring the origins of the ice floe that Polarstern was moored in for the MOSAiC expedition. Using a Lagrangian tracking technique in conjunction with a thermodynamic sea ice model, the authors establish the location of origin for the MOSAiC floe: North of the New Siberian Islands. The results described here will provide important context for future research based on MOSAiC. There are some places where the manuscript could be improved or clarified, but in general, these are minor. Occasionally, imprecise language is used – quantifying "a few" and "multiple" would be helpful, for instance. My main comment concerns the validation of the Lagrangian technique against satellite-derived floe positions. The computed trajectories appear to agree well with the satellite data (Figure 4a), but there is no measure of uncertainty in the satellite data included in this plot. Quantifying these uncertainties would be an important improvement. Additionally, neither the Lagrangian trajectories nor the satellite-derived positions have any indication of what time they were at each position in Figure 4a. Perhaps including a multicoloured trajectory line, with colour corresponding to day of year (and equivalent for the crosses) would aid the interpretation of this plot, and would give a better indication of how well the computed trajectories agree with the observations. Discussing how the discrepancy between computed trajectories and satellite observations evolves with time would provide important validation for the back-tracking results. Other places where I feel more explanation would strengthen the manuscript are detailed below in the specific comments section. I feel these comments are minor, and I look forward to seeing an updated version of the manuscript published in The Cryosphere.

Specific comments: Line 63: Have they really only been temporarily discontinued? With Arctic sea ice continuing to decline, surely the old style of drift stations have been permanently discontinued and replaced with new approaches like MOSAiC? Figure 1a: a box highlighting the region shown in Figure 1b would be helpful. Figure 1b: The term DNR is not mentioned until line 230 of the main text – as Figure 1 is discussed before this, better not to abbreviate in the caption, or to spell out what DNR stands for ∼line 70. Line 76: Key citations(s) for the 'previously described methods' would be useful here. Line 82: Which atmospheric reanalysis data? Line 90: Which historical forcing data? Line 100: A sentence or two briefly summarising the findings of the cited papers would be nice to better put into context what IceTrack has previously been used for. Line 107: Is the 25km grid high enough resolution to capture smaller scale circulation features (e.g. eddies)? If not, how are the results likely to be impacted by not including these? Line 109: Why 40%, rather than the 20% threshold used in Krumpen et al. (2019)? Line 125: How many ice categories / layers? Table 1: I assume the numbers in parentheses after the means are a measure of the uncertainty, but it isn't clear from the caption whether this is standard deviation, range, or some other measure. Clarification needed. Line 187-190: I appreciate that that unavailability of oceanographic results means that it is necessary to focus on atmospherically driven processes affecting the retreat of the ice edge in 2019. However, a short overview of the oceanic processes that play an important role in general (rather than specifically 2019) would help to better put this work into context. Line 204: In Figure 2a, there is a positive thickness anomaly immediately around the New Siberian Islands. The trend for the region in general is clearly negative, but 'negative thickness anomalies throughout the entire coastal zones' is phrased too strongly – the exception to the trend should at least be noted. Line 245: How many is "a few" days? It would be great if an estimate is possible, if not perhaps better to drop. Line 269: Given that only approximate locations of floes could be determined from the high (how high?) resolution satellite data, it would be good to see some measure of the uncertainty attached to the crosses shown in Figure 4a, or at least discussed in the text. Figure 4: Linked to my previous comment, it would also

be helpful to quantify how closely the satellite-derived positions match the computed trajectories. It is already clear from Figure 4a that they follow similar paths, but this doesn't necessarily show that they're in (approximately) the same place at the same time. Including a measure of the distance between the Xs and the corresponding point on the Lagrangian trajectories would answer this question, and would be interesting to compare to the (36 +/-20 km after 200 days) deviation found in previous work. Line 314: It would be interesting to include a brief overview of the 'not shown' analysis in the supplementary material. Line 318: How close was the ARK-12 cruise data "in the surroundings of the DNR" to the DNR?

Technical corrections: Line 187: I think a word is missing (e.g. "In the following section, . . .") Line 192: This reads a bit strangely. Perhaps "Ice dynamics and ice export in winter are important preconditioning mechanisms for the ice retreat in summer" would be better.